# YBX1 integration of oncogenic PI3K/mTOR signalling regulates the fitness of malignant epithelial cells

Yuchen Bai [1], Carolin Gotz[2,3], Ginevra Chincarini[1], Zixuan Zhao[4], Clare Slaney [1,5], Jarryd Boath[1], Luc Furic[1,5,6], Christopher Angel[7], Stephen M. Jane[8], Wayne A. Phillips [1,5], Steven A. Stacker [1,5], Camile S. Farah [9] & Charbel Darido [1,5] ✉

In heterogeneous head and neck cancer (HNC), subtype-specific treatment regimens are currently missing. An integrated analysis of patient HNC subtypes using single-cell sequencing and proteome profiles reveals an epithelial-mesenchymal transition (EMT) signature within the epithelial cancer-cell population. The EMT signature coincides with PI3K/mTOR inactivation in the mesenchymal subtype. Conversely, the signature is suppressed in epithelial cells of the basal subtype which exhibits hyperactive PI3K/mTOR signalling. We further identify YBX1 phosphorylation, downstream of the PI3K/mTOR pathway, restraining basal-like cancer cell proliferation. In contrast, YBX1 acts as a safeguard against the proliferation-to-invasion switch in mesenchymal-like epithelial cancer cells, and its loss accentuates partial-EMT and in vivo invasion. Interestingly, phospho-YBX1 that is mutually exclusive to partial-EMT, emerges as a prognostic marker for overall patient outcomes. These findings create a unique opportunity to sensitise mesenchymal cancer cells to PI3K/mTOR inhibitors by shifting them towards a basal-like subtype as a promising therapeutic approach against HNC.

Head and neck cancer (HNC) is the sixth most common cancer worldwide with a poor overall survival rate[1]. At initial diagnosis, more than 60% of HNC patients present with advanced-stage disease including local invasion, evidence of metastases to regional lymph nodes, or both[2]. HNC metastasis is a multi-step process that involves dissemination of cancer cells from the primary tumour site, nodal metastases, intravascular survival and circulation, seeding and propagation at a secondary organ[3]. Metastatic HNC is highly resistant to therapy with a five-year overall patient survival of less than 20% (approximately 10-months median overall survival)[4]. Therefore, understanding the molecular mechanisms of HNC metastasis is critical to enhance therapy response and reduce cancer morbidity and mortality[5].

HNC tumours have been classified into four major molecular subtypes: basal, mesenchymal, classical and atypical whereby the basal and mesenchymal subtypes represent more than 70% of HNC[6].

[1]Peter MacCallum Cancer Centre, 305 Grattan St, Melbourne, VIC 3000, Australia. [2]Department of Oral and Maxillofacial Surgery, Technische Universität München, Fakultät für Medizin, Klinikum rechts der Isar, Ismaningerstraße 22, 81675 Munich, Germany. [3]Department of Oral and Maxillofacial Surgery, Medizinische Universität Innsbruck, Anichstraße 35, 6020 Innsbruck, Austria. [4]Sun Yat-sen University Cancer Center, Yuexiu District, Guangzhou, Guangdong Province, China. [5]The Sir Peter MacCallum Department of Oncology, The University of Melbourne, Parkville, VIC 3010, Australia. [6]Cancer Program, Biomedicine Discovery Institute and Department of Anatomy and Developmental Biology, Monash University, Clayton, VIC 3800, Australia. [7]Department of Histopathology, Peter MacCallum Cancer Centre, Melbourne, VIC 3000, Australia. [8]Department of Medicine, Central Clinical School, Monash University, 99 Commercial Road, Melbourne, VIC 3004, Australia. [9]Australian Centre for Oral Oncology Research & Education; Fiona Stanley Hospital; Hollywood Private Hospital; Australian Clinical Labs, CQ University, Perth, WA 6009, Australia. ✉e-mail: charbel.darido@petermac.org

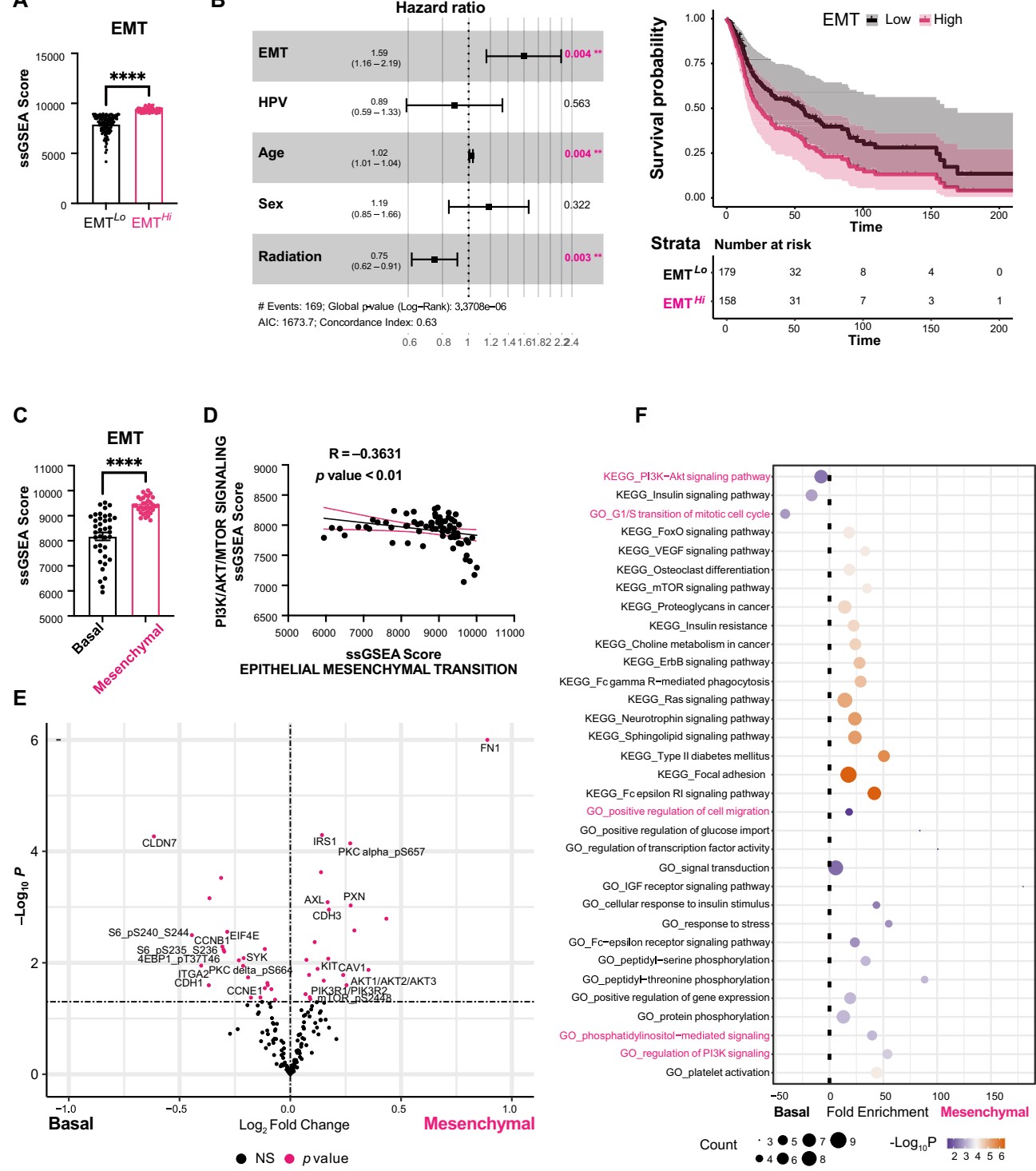

Tumours classified as mesenchymal reflect a basal transcriptomic signature in malignant epithelial cells with a large stromal component and were described as malignant-basal[7]. Invasive cells at the leading edge of combined malignant-basal HNC undergo partial epithelial to mesenchymal transition (p-EMT)[7]. While the classical and atypical HNC subtypes are able to metastasise at similar rates to the malignant-basal, they did not express p-EMT genes[7]. The p-EMT program reorganises the cell cytoskeleton to enable motogenesis at the expense of mitogenesis and manifests by decreased expression of epithelial markers (e.g. CDH1) and increased expression of mesenchymal markers (e.g. PDPN), required for HNC cell migratory and invasive abilities[8–10].

Moreover, in HNC patients, p-EMT serves as a predictive biomarker of nodal metastases, lymphovascular invasion and extranodal extension[3].

While proliferative and invasive cancer cells retain common genetic alterations, discrete molecular processes evolve in proliferating cancer cells to acquire an invasive potential[11–13]. However, molecular factors responsible for the proliferative to invasive switch remain poorly understood and represent an attractive, yet unexplored therapeutic opportunity against metastatic cancers.

Large-scale genomic and transcriptomic analyses of HNC revealed frequent dysregulation of the phosphatidylinositol 3-kinase (PI3K) signalling pathway components such as *EGFR*, *PIK3CA*, *AKT*, *PTEN* and

**Fig. 1 | The EMT signature inversely correlates with PI3K pathway activation in basal and mesenchymal HNC patient subtypes. A** Stratification of patient HNC based on the single-sample Gene Set Enrichment Analysis (ssGSEA) score of hallmark EMT genes using TCGA-HNC mRNA-seq data identified EMT[Lo] ($n = 127$) and EMT[Hi] ($n = 84$) patient groups. The comparison between the two groups was performed using an unpaired t test (**** two-tailed $p$-value < 0.0001). Data are shown as mean ± SEM. **B** Hazard ratio (HR) and adjusted Kaplan-Meier curve analyses of EMT for overall survival of $n = 337$ patients from the TCGA-HNC cohort. HRs were estimated using multivariate Cox proportional hazard models. Asterisks indicate statistical significance per clinically-relevant factor. HR estimates are shown as dots at the centre of error bars and error bars represent 95% confidence intervals of hazard ratios by two-sided Wald test. **C** The EMT ssGSEA score was significantly higher in mesenchymal ($n = 34$) compared to basal ($n = 38$) HNC subtype. The results are presented as mean ± SEM with **** two-tailed $p$-value < 0.0001. The comparison between the two groups was performed using an unpaired t test. **D** Significantly negative correlation between ssGSEA scores of the EMT and PI3K/AKT/mTOR gene sets from the basal and mesenchymal HNC ($n = 72$, Spearman's correlation, R = −0.3631, two-sided $p$-value=0.0017). 95% confidence bands of the best-fit line are shown in pink. **E** Volcano plot of differentially expressed proteins between basal and mesenchymal HNC samples using the TCGA-HNC reverse-phase protein array (RPPA) data. Labelled proteins associated with EMT were upregulated in the mesenchymal subtype and those associated with PI3K/AKT/mTOR were upregulated in the basal subtype. Significant differentially expressed proteins with two-sided $q$-value < 0.05 are highlighted in pink, NS (not significant). The comparison between the two groups was performed using an unpaired t test. **F** Gene Ontology (GO) and Kyoto Encyclopaedia of Genes and Genomes (KEGG) terms enrichment for significant DEGs using DAVID functional annotation bioinformatics microarray analysis. Basal HNC showed enrichment for PI3K/mTOR signalling and mitotic activation while mesenchymal samples were enriched for migratory activation. Statistical values were considered significant at a false discovery rate (FDR) with two-sided $p$-value < 0.01. Source data are provided as a Source Data file.

$mTOR$[14]. Over 40% of HNC patients present with amplification of the *PIK3CA* gene (encoding the catalytic subunit of the PI3K complex) and up to 10% with *PIK3CA* gain-of-function mutations[1,15,16]. However, compared to *PIK3CA* wild-type HNC patients, treatment of gain-of-function *PIK3CA*-mutant HNC patients with the PI3K inhibitor did not provide a therapeutic advantage[17]. These results suggest that other factors may influence PI3K signalling to confer therapy resistance in HNC.

The transcription/translation Y Box binding protein 1 (YBX1) is a prognostic biomarker for disease-specific survival in HNC and is significantly increased in high-grade HNC cells at the tumour invasive front[18]. YBX1 which plays a crucial, albeit controversial, role in cell proliferation and invasion was shown to transcriptionally promotes cell proliferation in the nucleus[19] and to prevent protein translation in the cytoplasm[20]. In this study, we investigate the relationship between the PI3K signalling and YBX1 and their roles in HNC cell proliferation and invasion. Our data identify HNC subtype-specific, distinct functions for YBX1 and PI3K-dependent phospho-YBX1 in the regulation of the proliferative to invasive switch, with therapeutic implications against metastatic HNC.

## Results
### The EMT signature inversely correlates with PI3K pathway activation in basal and mesenchymal HNC patient subtypes
To evaluate the EMT signature score in patient HNC, single-sample gene set enrichment analysis (ssGSEA) was performed by applying hallmark gene sets to The Cancer Genome Atlas Head-Neck Squamous Cell Carcinoma (TCGA-HNC) cohort (Fig. S1A). We identified two distinct subgroups of HNC based on EMT gene expression, EMT[Hi] and EMT[Lo] (Fig. 1A). ESTIMATE and Stroma scores on bulk RNA-seq show a biased EMT signature in the mesenchymal subtype (Fig. S1B) with the overall survival, disease-specific survival, and progression-free survival are significantly poorer in EMT[Hi] compared to EMT[Lo] patients (Fig. S1C). We further conducted multivariate Cox regression survival analyses to adjust for clinically-relevant risk factors including HPV status, age, sex, and treatment (i.e., radiotherapy). Importantly, the forest plot for overall survival hazard ratios indicates that the EMT signature is an independent indicator of survival outcomes after adjustment (Fig. 1B). Additionally, mesenchymal tumours displayed a significantly higher EMT score, compared to the basal subtype (Fig. 1C). We further assessed whether any correlation exists between EMT and intracellular signalling pathways in HNC tumours using TCGA-HNC transcriptomic data (Fig. S1D). A dynamic range of correlations appeared but importantly, a significant negative correlation (−0.3631, $p$-value < 0.01) was obtained between EMT and PI3K/AKT/mTOR signalling (Fig. 1D). This was further confirmed by differentially expressed proteins using the TCGA-HNC reverse phase protein array (RPPA) data (Fig. 1E). Interestingly, the G1/S mitotic cell cycle transition pathway was enriched in the basal subtype while positive regulation of cell migration was associated with the mesenchymal subtype (Fig. 1F). Enrichment for the PI3K signalling pathway was consistent in both subtypes suggesting an important regulation of this pathway by activators and inhibitors. These data indicate that PI3K/mTOR signalling is active in proliferative cells of the basal HNC subtype while an EMT signature in the mesenchymal HNC subtype correlates with the inhibition of PI3K/mTOR signalling, decrease in cell proliferation, induction of cell migration, and a poor prognosis.

### Single cell analysis of patient HNC uncovered an epithelial-specific inverse correlation between partial EMT and PI3K signalling
To accurately evaluate the EMT status of malignant epithelial cells and its correlation with PI3K signalling outside the biased bulk RNA-seq (Fig. S1B), we explored transcriptomic heterogeneity at a cellular resolution using single-cell RNA sequencing (scRNA-seq). scRNA-seq profiles of primary and metastatic tumours were generated from four treatment-naïve patients (GSE140042) (Fig. 2A, B). The single-cell transcriptomes from 12,341 cells were retained after initial quality controls and partitioned into 13 clusters by gene expression levels. Individual clusters were then annotated according to characteristic marker genes (Fig. 2C–E). Malignant epithelial cells (cluster 4) were identified within the cellular heterogeneity of the tumour using epithelial-specific markers (Fig. 2F). A finer TCGA-HNC subtype classification (Fig. 2G) depicted intra-cellular heterogeneity within the malignant epithelial cells, demonstrating a dominant basal subtype signature. Within the malignant epithelial cells, we observed enrichment for the EMT signature in G1-arrested cancer cells while the PI3K/AKT/mTOR signalling was active in cycling cells (S and G2/M phases) (Fig. 2H). This agrees with the TCGA-HNC bulk RNA sequencing data in which the EMT signature negatively correlates with PI3K-active cycling cells (Fig. 1D). Notably, signatures from these cells differ between primary tumours and matched lymph node metastases (Fig. 2G); lymph node metastatic cells characteristically lacked the mesenchymal gene signature. Evaluation of the p-EMT program identified key p-EMT genes including extracellular matrix (PDPN), mesenchymal markers and EMT regulators in metastatic epithelial cells (Fig. 2I, J). Moreover, cells with a p-EMT[Hi] signature had a decreased proliferation rate (Fig. 2K). We validated the findings using two independent scRNA-seq datasets (GSE103322: 2215 tumour cells from 7 patients and GSE164690: 13875 tumour cells from 11 patients) to confirm the inversed correlation between EMT and PI3K-related cell proliferation. The cell cycle analysis shows enrichment for the EMT signature in G1-arrested cancer cells while the PI3K-AKT-mTOR signalling was active in cycling cells (Fig. S2), consistent with the scRNA-seq analysis of patient samples (GSE140042) and bulk RNA-seq studies. Taken together, our scRNA-seq analysis of epithelial HNC cells identified a p-EMT state that inversely correlates with PI3K/mTOR signalling and cell cycling at a single epithelial cell level.

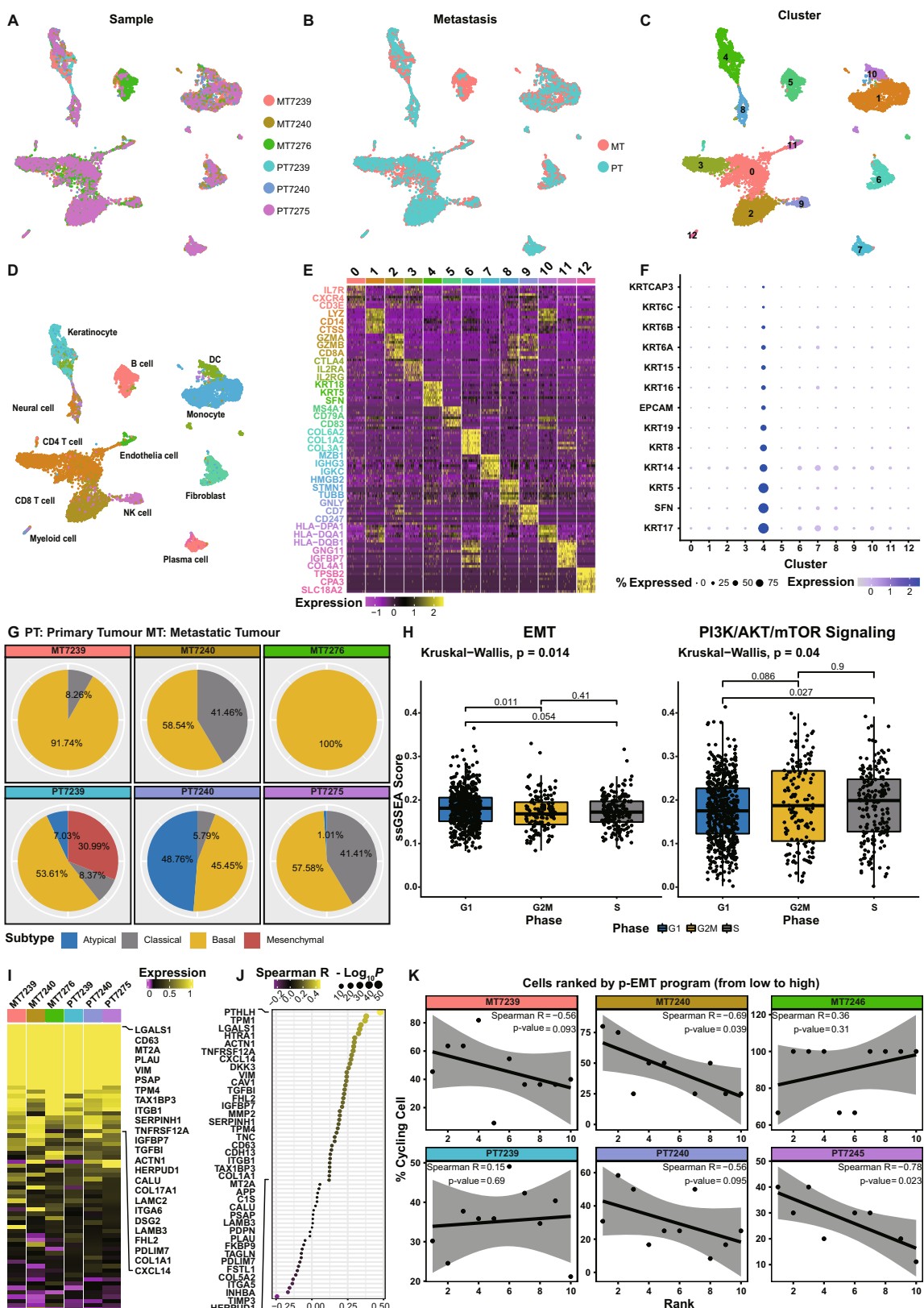

### The YBX1 factor links the PI3K signalling to partial EMT in specific HNC subtypes

Six HNC cell lines were classified into molecular subtypes using the transcriptomic signature of subtypes identified in the TCGA-HNC and the Cancer Cell Line Encyclopaedia (CCLE) databases[21]. SCC15 and SCC25 corresponded to the mesenchymal and basal like

subtypes, respectively, and showed the highest correlation coefficients among the cell lines (Fig. 3A). Analysis of the PI3K pathway mutation in these cells revealed a missense mutation (putative driver) in the PI3K regulatory subunit (*PIK3R3*) gene in the mesenchymal-like SCC15 cells (Fig. 3B). Interestingly, *PIK3R3* is only found to be mutated in mesenchymal patient HNC (Fig. S3)[22].

**Fig. 2 | Single cell analysis of patient HNC uncovered an epithelial-specific opposite signatures for partial EMT and PI3K signalling. A**−**C** UMAP plot of single cells from 6 primary and metastatic patient tumours assigned to 13 clusters using the "uwot" package. MT: metastatic tumours collected from lymph nodes; PT: primary tumours. **D** The clusters were assigned to the indicated cell types by differentially expressed genes and presented as (**E**) heatmaps. Yellow: high expression; purple: low expression. Selected genes are highlighted and labelled in cluster-matching colours. **F** Dot plot shows gene expression of epithelial cell markers across the clusters containing 12,341 single cells that were sorted by average expression of these genes. Blue: high expression; gray: low expression; the size of the dots indicates the percentage of cells expressing specific genes. **G** Pie chart shows the percentage of primary (PT) and metastatic (MT) tumour cells matching atypical, classical, basal and metastatic subtype signatures. **H** Box plots indicating ssGSEA EMT scores were significantly higher in G1 compared to G2M/S phase indicating EMT is high in non-proliferating cells. Conversely, PI3K/AKT/mTOR

hallmark gene set scores were significantly higher in S phase indicating cell proliferation. Comparisons between groups (G1 = 587, G2M = 146, S = 191 cells) were performed using Kruskal-Wallis t test and unpaired two samples Wilcoxon test. Median values are shown in each boxplot. All box plots include the median line, the box denotes the interquartile range (IQR), whiskers denote the rest of the data distribution and outliers are denoted by points greater than ±1.5 × IQR. **I** Heatmap shows the expression of common p-EMT genes in metastatic and primary patient tumours. **J** Dot plot illustrates a positive Spearman correlation (two-sided *p*-value < 0.01) between the genes and p-EMT program in cancer cells. **K** Scatter graphs show a decreasing percentage of cycling malignant cells and increasing p-EMT ssGSEA scores. The malignant cells are divided into ten sliding windows for each tumour. The consistent negative correlation was calculated using Spearman's correlation test with two-sided *p*-value. 95% confidence bands of the best-fit line are shown in gray. Source data are provided as a Source Data file.

Additionally, basal-like SCC25 cells had no PI3K signalling related mutations.

To investigate the direct role of PI3K signalling in HNC, we developed a relevant genetic mouse model by breeding the conditional *Pik3ca*[H1047R] knock-in mouse[23] onto a *Grhl3*-deficient background[24,25] to induce the development of PI3K-dependent HNC; GRHL3 being a differentiation factor that prevents HNC[26,27] and is downregulated in HNC samples of the TCGA cohort, with a comparable level between epithelial cells of basal and non-basal subtypes in scRNA-seq (Fig. S4A). Knock-in of an *H1047R* mutation into one allele of the endogenous *Pik3ca* gene in keratin-14 positive epithelial cells using a constitutive K14-Cre recombinase did not show any head and neck phenotype. Interestingly, spontaneous tumours developed in *Pik3ca*[H1047R]*Grhl3*[cKO] double mutant mice in less than 3 months (Fig. S4B). Marked hyperactivation of PI3K signalling was observed in the tumours along with expression of the cell proliferation markers Ccnd1 and pMet (Fig. S4C), the epithelial marker Cdh1 and the phosphorylation of Ybx1, but not the p-EMT marker Pdpn (Fig. S4D). This data indicates that PIK3CA-driven tumour cell proliferation does not induce p-EMT.

Transcriptomic analysis of mesenchymal SCC15 cells demonstrated upregulation of mesenchymal markers and downregulation of PI3K pathway-related genes. In contrast, transcriptomic analysis of basal SCC25 cells indicated an increased mRNA expression of epithelial markers and upregulation of PI3K genes (Fig. 3C). These findings were validated by proteomic RPPA analyses showing hyperactivation of the PI3K signalling (upper panel) and expression of the proliferative markers (lower panel) in basal SCC25 cells (Fig. 3D) compared to the mesenchymal SCC15 cells which showed inactivation of the PI3K signalling and expression of the p-EMT marker PDPN (Fig. S5A). Interestingly, SCC15 and SCC25 show an opposite pattern of expression for the EMT markers N-cadherin and Twist1 (Fig. S5B).

To validate our findings in vivo, we established an orthotopic HNC xenograft mouse model by transplanting luciferase-tagged HNC cells into the tongue of immunocompromised mice. This approach recapitulated the location of primary tumours in patients[21] and allowed weekly monitoring of in vivo tumour progression using bioluminescence imaging over a 6-week period (Fig. 3E). Interestingly, engrafted basal-like cells developed in situ tumours within the tongue while mesenchymal-like cells grew invasive tumours that underwent regional metastasis to lymph nodes (Figure S5C). Additionally, CDH1 expression (epithelial marker) was observed in basal SCC25-derived tumours whereas mesenchymal SCC15 tumours were positive for PDPN (p-EMT marker) (Fig. 3F). The PI3K signalling components EGFR, EIF4E and pEIF4E were highly expressed in SCC25 xenografts compared to SCC15 (Figure S5D). This data indicates that the PI3K-active basal subtype favours tumour growth over invasion while p-EMT induction in the PI3K-inactive mesenchymal subtype confers a metastatic potential in vivo.

To identify the molecular switch linking PI3K inactivation to p-EMT induction, we investigated oncogenic factors that are known to regulate HNC cell proliferation and invasion. We prioritised the YBX1 factor whose overexpression and cytoplasmic localisation were identified at the invasive front of metastatic patients HNC[18]. Furthermore, YBX1 phosphorylation at Ser102 by the PI3K signalling was recently shown to positively correlate with the expression of EGFR to facilitate cell proliferation and tumour growth[19]. Hence, the expression and phosphorylation of YBX1 were assessed in subtype-specific HNC cells and xenografts. WB analyses showed the highest level of total YBX1 in mesenchymal SCC15 cells and phosphorylated forms of YBX1 in basal SCC25 cells within the HNC cell lines (Fig. 3G). Additionally, YBX1 was phosphorylated at Ser102 in response to 100 ng/ml of EGF or 20% FBS treatments of SCC25 and this phosphorylation was prevented by the dual PI3K/mTOR inhibitor BEZ235 (Figs. S5E and S5F). Immunofluorescence (IF) staining detected YBX1 mainly in the nuclear compartment of SCC25 whereas YBX1 was cytoplasmic in invasive SCC15 cells and phospho-YBX1 localised specifically to the mitosis spindles in dividing basal SCC25 cells (Fig. 3H). YBX1 and phospho-YBX1 IF quantification shows higher nuclear localisation in SCC25 compared to SCC15 (Fig. 3I) which was further validated using WB on cytoplasmic and nuclear cellular fractions following EGF treatment (Fig. S5G). This data demonstrates that nuclear YBX1 occurs in PI3K-active proliferative cells while its cytoplasmic counterpart associates with the PI3K-inactive invasive subtype.

**Loss of YBX1 inhibits cell proliferation in basal-like HNC cells with active PI3K signalling**
To evaluate the oncogenic function of YBX1 in the basal subtype, we employed a doxycycline (DOX)-induced CRISPR-Cas9 system with single guide RNAs against *YBX1*. Q-PCR and WB analyses were used to assess YBX1 knockdown efficiency (-80%) (Fig. 4A). Interestingly, YBX1 knockdown decreased the 3D-growth of SCC25 basal-like cells in an ultra-low attachment condition with a significant reduction in the number and size of colonies (Fig. 4B). Furthermore, diminished cell proliferation correlated with reduced SCC25 invasion in a transwell invasion assay (Fig. 4C). Whole transcriptome analysis of *YBX1*-knockdown SCC25 (−YBX1) compared to SCC25 parental (+YBX1) cells showed enrichment for genes involved in the suppression of G2/M checkpoint and ribosomal biogenesis, and activation of apoptosis and anti-tumour inflammatory responses (Fig. 4D). Importantly, YBX1 downregulation resulted in a significant loss of the E2F, YY1, and MYC/MAX transcription factor target genes. The gene signature which correlates with cell proliferation and survival was enriched in +YBX1 cells (Fig. 4E). Moreover, SCC25 −YBX1 cells treated with EGF showed strong activation of PI3K/mTOR signalling, suggesting that activation of PI3K signalling is a compensatory mechanism in basal-like cells (Fig. 4F).

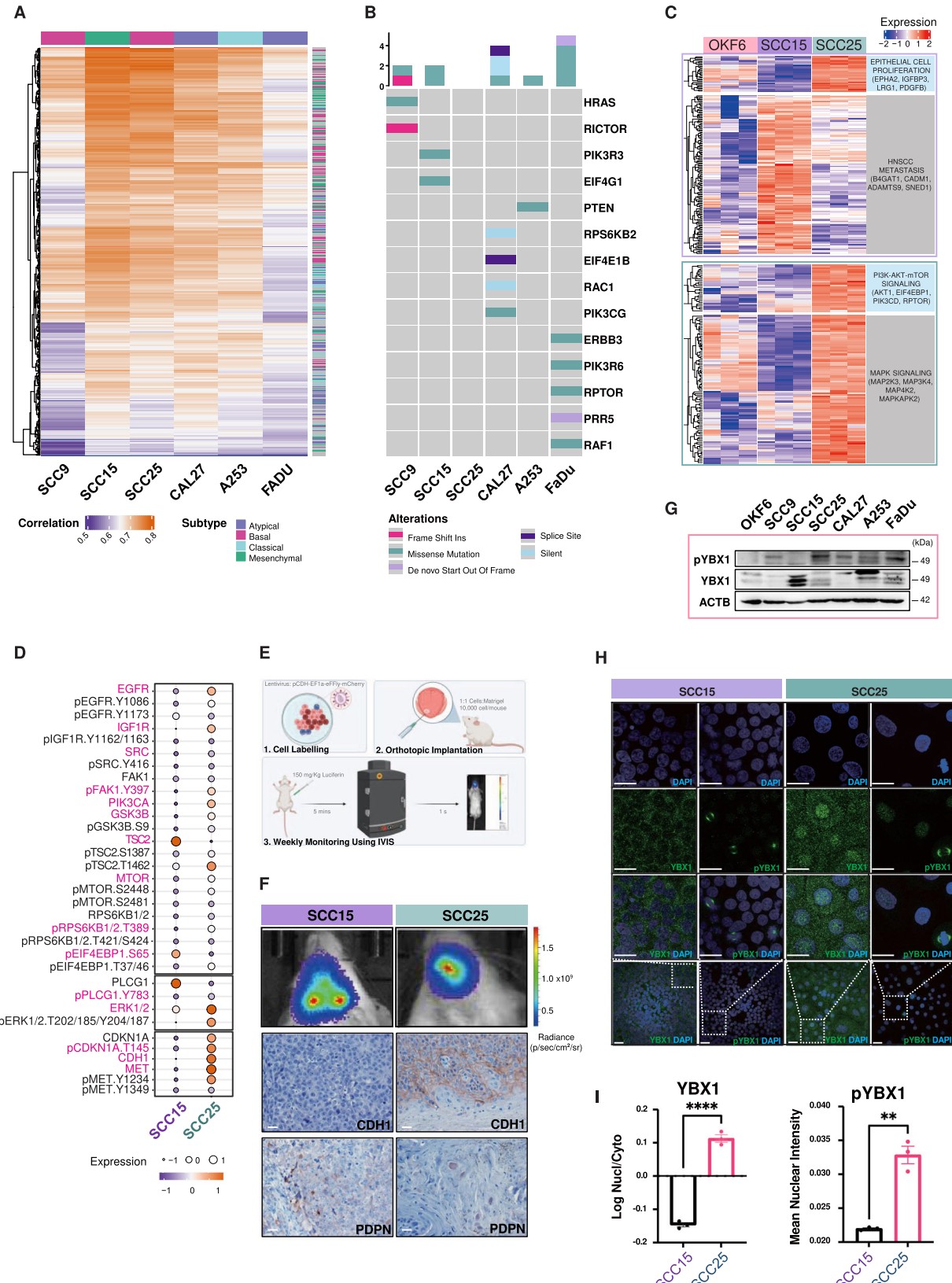

We next established orthotopic SCC25 (−/+YBX1) xenografts. YBX1 downregulation strikingly inhibited tumour growth as measured by bioluminescence imaging, the average intensity radiance and tumour weight (Fig. 4G). Residual tumours from SCC25 −YBX1 were confirmed negative for YBX1 by immunohistochemistry (IHC) staining (Fig. 4H). NSG mice bearing SCC25 (−/+YBX1) xenografts were administered daily the PI3K/mTOR inhibitor BEZ235 (35 mg/kg) by oral gavage over a 4-week period. Loss of YBX1 reduced tumour growth of SCC25 which was further reduced in response to BEZ235 treatment (Figs. S6A and S6B). Inactivation of the ribosomal protein S6 shown by loss of p-RPS6 compared to total RPS6 confirmed the efficacy of BEZ235 treatment in SCC25 xenografts (Fig. S6C). This data validates

**Fig. 3 | A subtype-specific negative correlation between partial EMT and PI3K signalling in vitro and in vivo. A** Heatmap of Spearman's correlations between HNC cell lines (x-axis) and TCGA-HNC primary tumours (y-axis). The HNC lines were classified as atypical (FADU and CAL27), classical (A253), basal (SCC9 and SCC25) and mesenchymal (SCC15) based on subtype-specific TCGA-HNC gene signatures. The subtype annotation bar is shown on top of the heatmap and on the right for TCGA-HNC primary tumours. **B** PI3K/AKT/mTOR signalling-related mutations were identified using CCLE and OncoPrint. **C** Heatmap showing mRNA expression of EMT markers and PI3K/AKT/mTOR signalling in normal-OKF6, basal-SCC25 and mesenchymal-SCC15 cells. **D** RPPA analyses of PI3K/AKT/mTOR signalling and EMT markers. Expression levels are shown as fluorescence intensity values for differentially expressed proteins (one-way ANOVA $p$-value < 0.05) and highlighted in pink. Circles and colours represent the level of expression for each protein. **E** Schematic illustration of orthotopic xenografts of HNC cells in NSG mouse tongues. Cells transduced with a lentivirus-containing luciferase gene were injected into the tongue and mice monitored weekly using PerkinElmer IVIS® Spectrum imaging system. **F** Representative images of bioluminescence at 38 days post-orthotropic implantation. IHC analysis of p-EMT markers in SCC15 and SCC25 xenografts. The epithelial marker CDH1 was expressed in SCC25 while the p-EMT marker PDPN was detected in SCC15. Magnification, X40; scale bars, 50 μm. **G** WB analysis of YBX1 and phospho-YBX1 in the HNC lines. β-actin was used as loading control. Data are representative of $n = 2$ biological replicates. **H** Immunofluorescence (IF) of YBX1 and p-YBX1 in SCC15 and SCC25. DAPI was used as counterstaining. Magnification, X40; scale bars, 50 μm. **I** Cytoplasmic and nuclear localisation of YBX1 and pYBX1 were quantitated using ImageJ from three independent experiments. YBX1 was localised in the cytoplasm of both lines with low nuclear detection of pYBX1 in mesenchymal SCC15 and high pYBX1 in the nucleus of basal SCC25 cells. Data are shown as mean ± SEM. The comparison between the two groups was performed using an unpaired t test (**$p$-value = 0.0011, ****$p$-value < 0.0001). Source data are provided as a Source Data file. Schematic Fig. 3E was created using Biorender.com.

the oncogenic function of YBX1 in promoting tumour development of basal HNC cells with active PI3K signalling.

## Cytoplasmic YBX1 induces p-EMT in the absence of PI3K signalling

The function of YBX1 was evaluated in mesenchymal-like SCC15 cells with inactive PI3K signalling. *YBX1* knockdown efficiency (-80%) in SCC15 (−YBX1) was determined by Q-PCR and WB (Fig. 5A). Compared to SCC15 (+YBX1), SCC15 (−YBX1) cells proliferated more and grew into bigger colonies in an ultra-low attachment condition (Fig. 5B) and were significantly more invasive in the transwell invasion assay (Fig. 5C). RNA sequencing of SCC15 −YBX1 demonstrated enrichment for genes related to PI3K/mTOR signalling and translation initiation (Fig. 5D). Furthermore, downregulation of negative PI3K regulators (e.g. TSC2) and upregulation of proliferation suppressors (e.g. CDKN1A) were observed in SCC15 −YBX1 cells, independent of EGF treatment (Fig. 5E), indicating a reduced PI3K-dependent proliferation.

Next, we established orthotopic SCC15 (−/+YBX1) xenografts to investigate the role of YBX1 in metastasis. Unlike SCC25, SCC15 −YBX1 xenografts exhibited a faster growth rate and a significantly higher metastatic potential compared to SCC15 (+YBX1) (Fig. 5F and Figs. S6A and S6B). However, tumour weight did not significantly differ between −YBX1 and +YBX1 xenografts (Fig. 5F). IHC and IF staining of YBX1 in GFP-labelled invasive SCC15 cells demonstrated YBX1 localisation to the invasive front of primary tumours (Fig. 5G). Moreover, YBX1 was strongly expressed in the lymph node metastatic lesions. NSG mice bearing SCC15 (−/+YBX1) xenografts were administered daily the PI3K/mTOR inhibitor BEZ235 (35 mg/kg) by oral gavage over a 4-week period. While loss of YBX1 induced a growth advantage for SCC15 xenografts, SCC15 −YBX1 developed therapy resistance to BEZ235 treatment with increased metastasis to lymph nodes (Figs. S6A and S6B) while p-RPS6 was completely loss (Fig. S6C). Taken together, these data indicate that YBX1 is a suppressor of metastasis in mesenchymal HNC with inactive PI3K signalling. Furthermore, we overexpressed a myc-tagged YBX1 in invasive SCC15 cells. Transfected cells (57.4%) were confirmed for YBX1 overexpression (OE) using flow cytometry (Fig. S7A) and by anti-MYC and anti-YBX1 western blots (Fig. S7B). The transwell invasion assay shows OE cells with nuclear YBX1 expression in green (IF for the MYC tag) are retained in the inner membrane (non-invasive) whether SCC15 cells with cytoplasmic YBX1 are detected in the outer membrane (invasive). The quantification of YBX1 OE demonstrates decreased nuclear to cytoplasmic ratio in invasive cells (Fig. S7C). While YBX1 is mainly shown in the cytoplasm of SCC15 cells (Fig. 3H), this data indicates that OE of YBX1 in SCC15 can localise to the nucleus to inhibit cell invasion.

## The PI3K-phospho-YBX1 axis in basal and mesenchymal HNC subtypes predicts patient prognosis

Considering the high rate of *PIK3CA* mutations in HNC, we evaluated the clinical significance of the PI3K-phospho-YBX1 signalling in patients. RPPA data from basal and mesenchymal HNC subtypes were extracted from the TCGA-HNC dataset ($n = 58$). Compared to basal HNC, mesenchymal tumours showed a lower YBX1 phosphorylation rate, but no significant difference in the expression of total YBX1 (Fig. 6A). In addition, phospho-YBX1 expression level, but not total YBX1, was found to be inversely correlated with the EMT score by a Spearman's correlation analysis (Fig. 6B). An unsupervised clustering of basal and mesenchymal patient HNC subtypes identified an inverse correlation of the PI3K-phospho-YBX1 axis with the EMT markers (Fig. 6C). Using the TCGA-HNC and matched RPPA data, expression of phospho-YBX1 predicted survival outcomes of HNC patients from all subtypes ($n = 337$). Overall survival rates of patients with high expression of phospho-YBX1 (p-YBX1$^{Hi}$) were significantly and independently improved when compared to p-YBX1$^{Lo}$ patients (Fig. 6D). In addition, the p-YBX1$^{Hi}$ HNC tumours were more differentiated with lower histological grade, disease stage and lymph node involvement compared to p-YBX1$^{Lo}$ HNC patients (Fig. 6E). It should be noted that total YBX1 expression did not significantly differ between p-YBX1$^{Lo}$ and p-YBX1$^{Hi}$ tumours (Fig. 6F), underscoring the impact of YBX1 phosphorylation on patient outcomes. Furthermore, a detailed analysis of p-YBX1$^{Hi}$ tumours identified upregulation of the PI3K signalling pathway as shown in the volcano plot (Fig. 6G). The data was validated in patient HNC tissue microarrays by IHC staining and demonstrated a mutually exclusive expression pattern for phospho-YBX1 and PDPN, particularly at the invasive front of HNC (Fig. 6H and Figs. S8A–C). Overall, our data establish the PI3K-phospho-YBX1 axis as an oncogenic driver of tumour growth in patient HNC basal subtype and the YBX1 factor as a suppressor of metastasis in the mesenchymal subtype (Fig. 7).

## Discussion

Previously, heterogeneous HNCs have been stratified into four distinct gene-expression subtypes[28]. Basal, mesenchymal, atypical and classical subtypes were characterised by somatic mutations, copy number alterations, gene expression and DNA methylation profiles[15,29]. Recently, HNC tumours were suggested to be refined into three subtypes (malignant-basal, classical, and atypical), with the mesenchymal subtype reflecting malignant-basal tumours with a large stromal component[7]. Our current data agrees with the HNC malignant-basal subtyping analysis and further adds a PI3K/mTOR-dependent sub-classification for potential targeted therapy stratification.

Mutations in components of the PI3K pathway are among the most frequent in HNC[22] with an increased burden in metastatic disease[5]. The clinical trials using PI3K signalling inhibitors have shown substantial differences in therapy response against solid cancers including HNC[14,30–32]. In line with this, our data indicate that the mesenchymal subtype is molecularly programmed to endure PI3K/mTOR inhibition and suggest the inclusion of HNC molecular

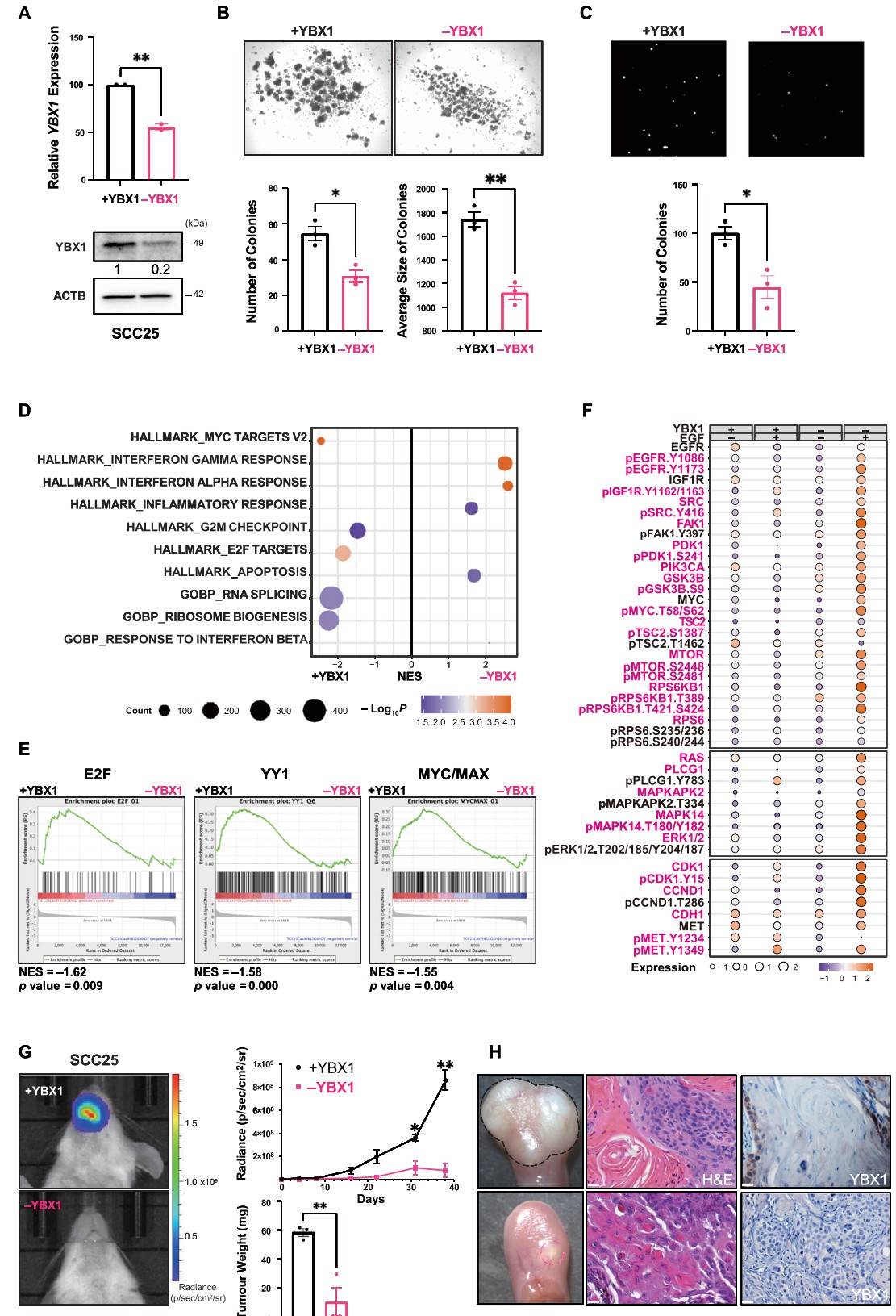

subtyping in future trials to predict therapy response. Moreover, since basal-like SCC25 cells are highly sensitive to multiple PI3K inhibitors[33], chemotherapy (Cisplatin/5-FU)[34,35] and Cetuximab[36], this study provides mechanistic insights into therapy resistance[37] and the downstream effectors in the PI3K-inactive mesenchymal subtype. Considering the differential response of basal and mesenchymal

subtypes to Cisplatin/5-FU, HNC patient of the basal subtype are proposed to respond better than those of the mesenchymal subtype. Our data are also relevant to the induction of EMT in resistant Cetuximab-treated HNC in the clinic[38]. Because the proliferation-invasion switch is the basis of therapy resistance, metastasis and cancer mortality[39,40], we showed phosphorylation of YBX1 in basal-like HNC cells with active

**Fig. 4 | Loss of YBX1 inhibits cell proliferation in basal-like HNC cells with active PI3K signalling. A** CRISPR-Cas9 DOX-induced (1 µM, 72 hours) YBX1-knockdown in SCC25 resulted in ~80% downregulation (** two-sided *p*-value=0.0052) of YBX1 by Q-PCR (upper panel) and WB (lower panel). WB images were quantified using ImageJ (*n* = 2 independent experiments). **B** Compared to SCC25 (+ YBX1), SCC25 (−YBX1) grew fewer (* *p*-value=0.01) and smaller colonies (** *p*-value=0.0016) on ultra-low attachment plates for seven days. Images of spheroids were captured using an EVOS cell-imaging microscope. The number and average size of colonies were quantified using ImageJ (*n* = 3 experiments). **C** Transwell invasion of SCC25 (−YBX1) cells (*n* = 3 experiments). The number of invading cells was quantified using ImageJ. Magnification, X40; scale bars, 500 µm, * *p*-value = 0.0144. **D** GO and KEGG term enrichments for significant DEGs showed enrichment for genes involved in G2/M checkpoint suppression and ribosomal biogenesis, activation of apoptosis and anti-tumour inflammatory responses. **E** Significant loss of E2F, YY1 and MYC/MAX target genes in SCC25 (−YBX1) using GSEA regulatory target gene sets. The adjusted significance was empirically determined by 1000 gene-set permutations. **F** RPPA analyses of PI3K/AKT/mTOR signalling and EMT markers in SCC25 (−/+YBX1) treated with EGF (100 ng/mL) for 30 mins. Expression levels are shown as fluorescence intensity values for differentially expressed proteins (one-way ANOVA *p*-value < 0.05) and highlighted in pink. **G** Bioluminescence imaging of SCC25 −YBX1 (pink) and SCC25 + YBX1 (black) tumours (*n* = 3 experiments, 3 mice/group/experiments) at 38 days post-implantation. The radiance of bioluminescence was measured weekly, and data shown as mean ± SEM at each timepoint (* *p*-value = 0.0179, ** *p*-value = 0.0018). Tumours were weighed at sacrifice, and data shown as mean ± SEM (** *p*-value = 0.0085). **H** Optical images showing significant growth inhibition of SCC25 (−YBX1) xenografts and H&E staining of undifferentiated SCC25 (+ YBX1) compared to SCC25 (−YBX1). IHC confirmed loss of YBX1 in SCC25 (−YBX1) tumours. Magnification, X40; scale bars, 50 µm. (**A**−**C**) and (**G**) data are shown as mean ± SEM. The comparison between the two groups was performed using an unpaired two-sided t test (**p*-value < 0.05, ***p*-value < 0.01, *****p*-value < 0.0001). Source data are provided as a Source Data file.

PI3K signalling confers an oncogenic role for this factor, while unphosphorylated YBX1 acts as a suppressor of metastasis in mesenchymal-like cells with inactive PI3K signalling. This subtype-specific function of YBX1 is PI3K-dependent and underscores a pivotal role for YBX1 in fine-tuning the cancer cell fitness with a critical impact on HNC patient outcomes. Genetic alterations are rarely detected at the *YBX1* locus, with only 2 amplifications identified in 2 HNC samples and a missense mutation (R279) in 1 sample (TCGA-HNC patient database, *n* = 529) (Fig. S8D). Moreover, YBX1 was shown to be regulated at the post-translational level as the main determinant of its function[41]. PI3K-mediated phosphorylation of YBX1 at Ser102 induces its translocation to the nucleus and results in the transcription of E2F, YY1 and MYC/MAX proliferative target genes (Figs. 4D, E)[41,42]. It remains to be tested whether other phosphorylations of YBX1 such as phospho-S209[43] affect the YBX1 function in tumour growth and invasion[44], and in *PIK3CA* mutant HNC as seen in *JAK2* mutant hematopoietic cancers[45]. In the absence of PI3K signalling, YBX1 binds to mRNAs of p-EMT markers in the cytoplasm[20]. Reduced levels of cytoplasmic YBX1 promote translation of p-EMT transcripts while high levels compete with eIF4E to inhibit the initiation of translation[46]. This function is evident in the mesenchymal-like SCC15 with inactive PI3K/mTOR/eIF4E and expression of cytoplasmic YBX1. Interestingly, *YBX1* knockdown in these cells activates the initiation of translation and results in increased colony growth and invasion. Therefore, YBX1-dependent translational regulation of p-EMT factors constitutes a vulnerability for therapeutic reversion of the metastatic phenotype[47].

Our data highlights the role of epithelial factors as pre-dominant regulators of the EMT switch and cancer cell fitness. It is equally important to consider the role of the tumour micro-environment in future studies, particularly the interactions between immune and malignant epithelial cells through secreted proteins[48,49]. Moreover, the immune checkpoint PD-1/PD-L1 inhibitors Nivolumab and Pembrolizumab are FDA-approved for patients with recurrent or metastatic HNC and, interestingly, YBX1 has been shown to regulate the expression of PD-L1 within an immunosuppressive microenvironment[50], establishing this factor as an important therapeutic target against both epithelial cancer cells and the tumour microenvironment.

In conclusion, this study discovered a mutually exclusive interplay between PI3K-mediated cell proliferation and p-EMT-initiated HNC invasion in basal and mesenchymal HNC subtypes and identified PI3K-dependent phosphorylation of YBX1 as a limiting factor of the switch to invasion. Our data also associate the subtype-specific PI3K-phospho-YBX1 axis with the patient survival outcomes. Future studies aiming at activating phospho-YBX1 may prove effective in shifting the mesenchymal phenotype towards a basal subtype and would constitute a promising therapeutic avenue against metastatic HNC.

## Methods

### RNA-seq analysis

The total RNA quantity was measured using Qubit RNA HS (Thermo Fisher Scientific). 500 ng of total RNA was used for library preparation according to standard protocols (QuantSeq 3′mRNA-seq FWD, Lexogen). Indexed libraries were pooled and sequenced on a NextSeq500 (Illumina). A total of 5–15 million single-end 75 bp reads were generated per sample. Sequence reads were trimmed and aligned to hg38 genome using Cutadapt and HISAT2 packages. Gene counts were obtained from featureCounts. Expression normalisation and filtration were performed using Limma. All analysis packages were operated within the Galaxy suite environment (version 4.0). Gene set enrichment analysis (GSEA_v4.3.2) was used for functional enrichment analysis of annotated terms from GO, KEGG and Hallmark using the human genome (GRCh38) as background.

### Bioinformatic datasets and analyses

Publicly available datasets were retrieved from TCGA Research Network using cBioportal (cbioportal.org). Gene and protein expression and patient survival information for 337 HNC samples were analysed. Sample purity was calculated using the tumour purity algorithm ESTIMATE. Gene mutation and expression profiling of the cell lines were retrieved from CCLE using depmap portal (depmap.org). Publicly available scRNA-seq data ("GSE140042"), ("GSE103322"), ("GSE164690") were retrieved from Gene Expression Omnibus (GEO) (ncbi.nlm.nih.gov/geo). Single-sample gene set enrichment analysis (ssGSEA) module on GenePattern (cloud.genepattern.org) was used with EMT and PI3K gene sets to generate the EMT and PI3K signature scores (HALLMARK_EPITHELIAL_MESENCHYMAL_TRANSITION, HALLMARK_PI3K_AKT_MTOR_SIGNALING).

### Single cell analysis

Droplet-based scRNA-seq data (GSE140042 and GSE164690) on the 10X Genomics Chromium platform were processed in RStudio using the Seurat package (version 4.1.0). Only HNC samples with 2000–4000 sequenced cells were considered for downstream analyses while others were excluded. Cells with more than 200 RNA features were retained and RNA features detected in more than 2 cells were considered for scRNA-seq analyses. Low-quality barcodes and unique molecular identifiers (UMIs) were filtered, mapped to human genome (GRCh38) and batch normalised using the Cell Ranger pipeline v3.0.2 (10X Genomics). The cell-gene matrix of UMI counts was then imported to Seurat. Genes expressed in >2 cells and cells with at least 200 genes were retained. Seurat objects were subsequently normalised, scaled, and integrated using CCA algorithm. A dimensional reduction matrix and clustering were obtained by aligning the CCA subspaces. Differentially expressed genes in individual clusters were identified and used for cell type assignment in each cluster by the

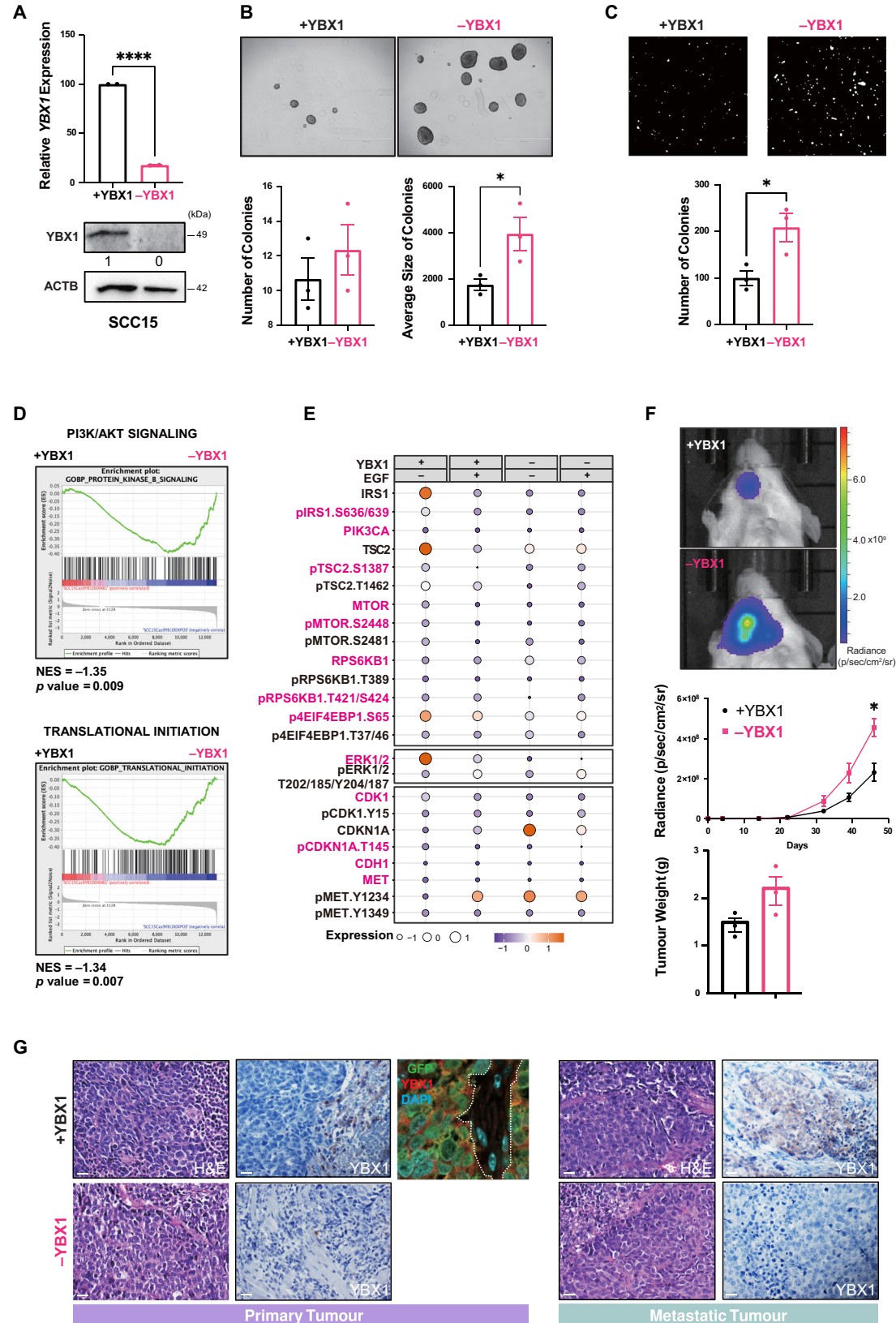

CellMarker human cell markers. The cell-cycle score was calculated based on the expression of S and G2/M phase markers, and cells expressing neither marker were identified in the G1 phase. EMT, p-EMT and PI3K ssGSEA scores were achieved using escape (version 1.3.3). For TCGA subtype assignment, differentially expressed genes (log fold-change > 2 and $p < 0.01$ by limma) in each of the four subtypes were identified, and single cells were scored by the four subtype gene sets using SingleR (version 1.8.1).

## HNC cell lines and cell culture
The normal oral epithelial cell line OKF6 was purchased from the Harvard Skin Disease Research Centre (HSDRC, Boston MA). The oral

**Fig. 5 | Cytoplasmic YBX1 induces p-EMT in the absence of PI3K signalling.**
**A** CRISPR-Cas9 DOX-induced (1 μM, 72 hours) YBX1-knockdown in SCC15 cells resulted in ~80% downregulation (****p-value < 0.0001) of YBX1 by Q-PCR (upper panel) and WB (lower panel). WB images were quantified using ImageJ (n = 2 independent experiments) **B** Compared to SCC15 (+ YBX1), SCC15 (−YBX1) cells grew more and bigger colonies (* p-value = 0.0448) on ultra-low attachment plates for seven days. Images of spheroids were captured using an EVOS cell imaging microscope. The number and average size of colonies were quantitated using the ImageJ (n = 3 experiments). **C** Transwell invasion of SCC15 (−YBX1) cells (n = 3 experiments). The number of invading cells was quantitated using ImageJ. Magnification, X40; scale bars, 500 μm, * p-value = 0.0321. **D** Significant enrichments for the protein kinase B signalling and the translation initiation in SCC15 (−YBX1) cells using GO gene sets. The significance was empirically determined by 1000 gene-set permutations. **E** RPPA analyses of PI3K/AKT/mTOR signalling and

EMT markers in SCC15 (−/+ YBX1) treated with EGF (100 ng/mL) for 30 mins. Expression levels are shown as relative fluorescence intensity values for differentially expressed proteins (one-way ANOVA p-value< 0.05) and highlighted in pink. **F** Bioluminescence imaging of live animals injected with either SCC15 (−YBX1) or SCC15 (+ YBX1) (n = 3 experiments, 3 mice/group/experiments) at 38 days post-transplantation. The radiance of bioluminescence was measured weekly, and data shown as mean ± SEM at each timepoint (*p-value = 0.0247). Tumours were weighed at sacrifice, and data shown as mean ± SEM.
**G** Acceleration of orthotopic tumour growth for SCC15 (−YBX1) confirmed by H&E staining of tumours and IHC showing loss of YBX1 in primary and metastatic tumours compared to SCC15 (+ YBX1). Magnification, X40; scale bars, 50μm.
**(A–C)** and **(F)** data are shown as mean ± SEM. The comparison between the two groups was performed using an unpaired two-sided t test (*p-value < 0.05, **p-value < 0.01, ****p-value < 0.0001). Source data are provided as a Source Data file.

cancer cell lines SCC9 (CRL-1629), SCC15 (CRL-1623), SCC25 (CRL-1628), and CAL27 (CRL-2095), A253 (HTB-41) and FaDu (HTB-43) were purchased from the American Type Culture Collection (ATCC, Manassas, VA). All cell lines were authenticated and validated by short tandem repeat (STR) profiling and tested negative for mycoplasma contamination[33]. OKF6 and SCC25 were cultured in keratinocyte serum-free medium (K-SFM, Gibco™) supplemented with growth factors (25 μg/mL BPE, 0.2 ng/mL EGF and 0.3 mM CaCl₂) and 1% penicillin-streptomycin (P/S). SCC9, SCC15, and CAL27 cells were cultured in Dulbecco's modified Eagle's medium (DMEM, Gibco™) with 10% foetal bovine serum (FBS) and 1% P/S. All cell lines were cultured at 37 °C in a 5% $CO_2$ humidified incubator and maintained at less than 25 passages. To activate the PI3K signalling, cells were grown in normal media and treated with EGF (100 ng/mL) for 30 minutes. To inhibit the PI3K signalling, cells were grown in normal media and treated with 100 nM BEZ235 (HY-50673, MedChemExpress) for 6 hours. The media was removed, and cells were washed with phosphate-buffered saline (PBS), then lysed in RIPA buffer for western blot analyses.

### Generation of lentiviral-transduced stable cell lines
Lentiviral expression constructs were mixed with the packaging plasmids pMDLgRRE, pRSV-Rev, pCMV-VSVG (Addgene) and transfected into lenti-X293T cells (Takara Bio, #632180) using Lipofectamine 2000 (Thermo Fisher Scientific) according to the manufacturer's protocol. Virus-containing conditioned media was collected 48–72 h after transfection and filtered using a 0.45 μm filter. Viral conditioned media was supplemented with polybrene and applied to sub-confluent cells. Conditioned media was replaced with fresh cell culture media 6 h post transduction. Double-positive GFP and mCherry-expressing cells were sorted using a FACS Calibur cell sorter 72 h post transduction and then expanded in culture. Doxycycline (Dox) Hyclate (Sigma Aldrich) was added (1 μg/ml) to the culture media to induce CRISPR-Cas9-mediated gene deletion for downstream analysis. Gifted plasmids were used for the inducible sgRNA construct pFgh1UTG and the FUCas9Cherry. The sgRNA targeting sequences for YBX1 (5′-GTAATGGCTTTTGTAGGGTG-3′ and 5′-GTTTGACACCGTTCATTGCA-3) were designed using the web-based tool CHOPCHOP (http://chopchop.cbu.unib.no/). The knockout efficiency was validated by quantitative polymerase chain reaction (Q-PCR) using human YBX1 primers (forward: 5′-AAGTGATGGAGGG TGCTGAC-3′ and reverse: 5′- TTCTTCATTGCCGTCCTCTC-3′) and housekeeping GAPDH primers (forward:5′- ACCCAGAAGACTGTGGAT GG −3′ and reverse: 5′- CAGTGAGCTTCCCGTTCAG −3′).

### Clonogenicity and invasion assays
Clonogenicity was assayed by seeding 500 cells of SCC15 and SCC25 at day 0 onto ultra-low attachment plates (Corning) and cultured in normal growth media. After 7 days, the images of spheroids were recorded on an EVOS live cell imaging microscope. The number of colonies was counted using Image J (version1.53). For the invasion assay, 24-well transwell plates with 8 μm pore size (Corning Costar,

USA) were used. The polycarbonate membrane was coated with Matrigel (Corning, USA) diluted in a coating buffer at 1:9 ratio and allowed to set for 2 hours. Next, cells were starved for 24 hours and seeded at 100,000 cells per Matrigel-coated well in a serum-free medium (top chamber) while complete culture media was added into the bottom chamber. The cells were incubated for 25 h at 37 °C in a humidified 5%$CO_2$ incubator and then fixed in 4% formalin at 4 °C overnight. Invasive cells in the transwell membranes were permeabilised with triton X-100 and stained by 1 ng/mL DAPI dye in methanol. Images were acquired using a BX-51 Olympus microscope and SPOT software 5.0.

### Protein extraction, western blotting (WB) and immunohistochemistry (IHC)
Cells were lysed in RIPA buffer with 1X protease inhibitor (Roche), and protein concentrations were determined using the DC protein assay kit (Bio-Rad) against BSA (Bovine serum albumin) standards according to the manufacturer's protocol. Absorbance readings were made on a Cytation 3 cell imaging multi-mode reader (Agilent, USA). For WB analysis, 30 μg total protein was denatured (95 °C, 2 min) in Laemmli Buffer and resolved on 10% polyacrylamide gels. Proteins were transferred to PVDF membranes (Millipore), blocked in 5% slim milk powder or BSA in TBS-T (Tris Buffered Saline with Tween) buffer for 2 hours. Membranes were then incubated with the primary antibodies listed in Supplementary Table 1 in TBS-T with 5% (w/v) BSA or 5% (w/v) milk powder overnight at 4 °C. Membranes were then washed in TBS-T and incubated with HRP-conjugated secondary antibodies. Membranes were washed and developed using Clarity Western ECL Blotting Substrate (Bio-Rad) and imaged on a Chemi-Doc MP Imaging System (Bio-Rad). β-actin and tubulin were used as protein loading controls. For IHC, whole mouse tongues were fixed in formalin and paraffin embedded. Tumour sections (5 μM) were deparaffinized, antigen was retrieved in sodium citrate and the endogenous peroxidase activity was blocked using 3% $H_2O_2$. The sections were then blocked with 5% normal goat serum and incubated with primary antibodies overnight followed by biotin-linked secondary antibodies incubated for 1 hour at room temperature. Chromogenic colour was developed using 3,3-diaminobenzidine (DAB). Slides were then counter-stained in Hematoxylin and a coverslip was mounted. No primary or no secondary antibodies were included in our IHC experiments as controls for the antibody staining specificity. Knockout cells were additional negative controls to validate the YBX1 antibody specificity.

The patient HNC tissue arrays contained 123 patient tumours that were stained in serial sections for phospho-YBX1 and PDPN. All microscopy images were acquired using a BX-51 Olympus microscope and SPOT software 5.0. The semi-quantification of the nuclear or cytoplasmic localisation in SCC15 and SCC25 was performed using CellProfiler 4.2.4. The semi-quantification of pYBX1 and PDPN region of interests (ROIs) in serial sections of the HNC tissue arrays was

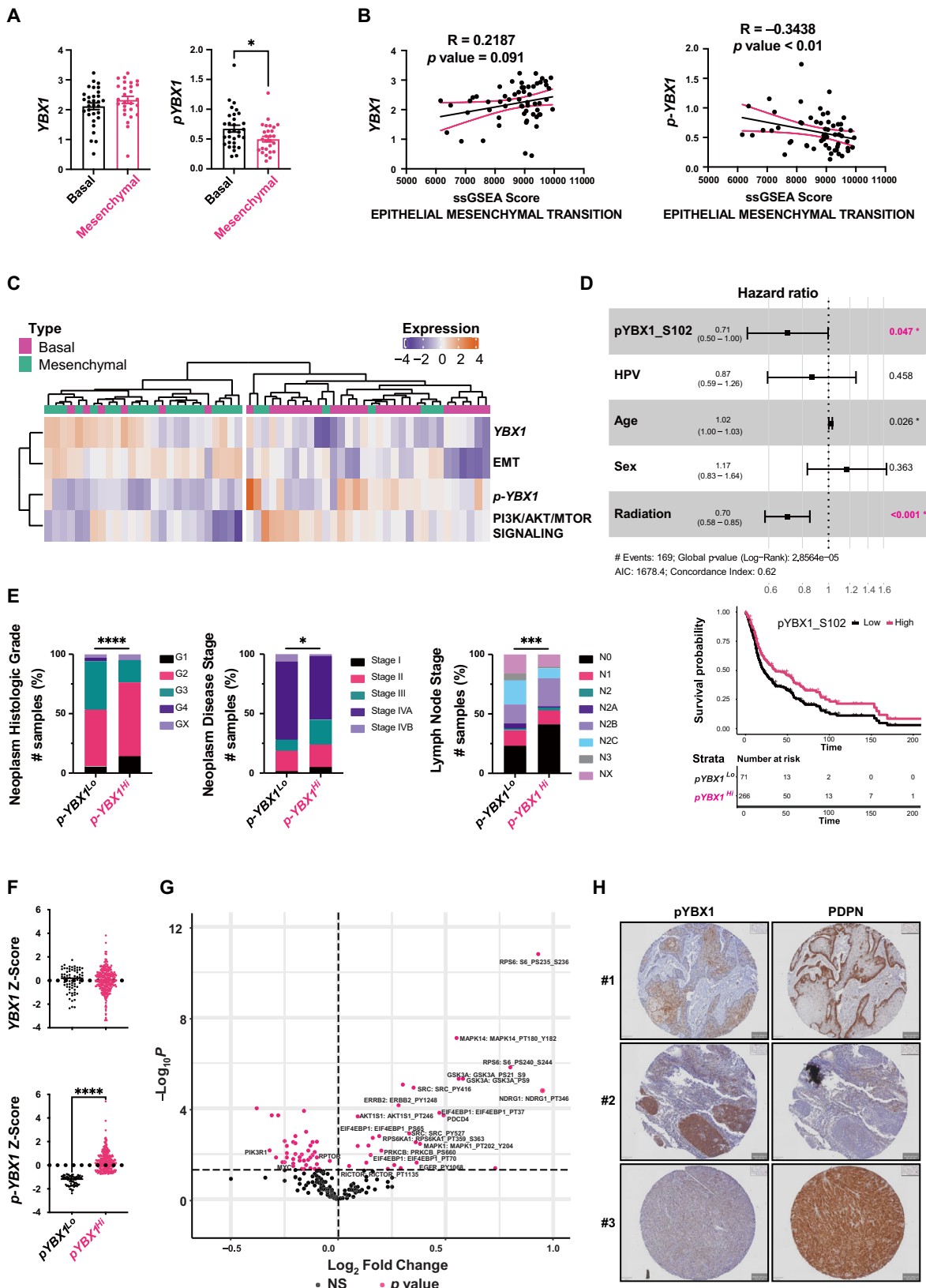

analysed on the HALO quantitative image analysis platform version 2.0 (Indica Labs) using random forest.

**Genetic mouse models and orthotopic xenografts**

All animal studies including breeding, experiments and euthanasia protocols were performed in accordance with the National Code of Practice for the Care and Use of Animals for Scientific Purposes and the Peter MacCallum Cancer Centre Animal Ethics guidelines and were approved by the Institutional Animal Experimentation Ethics Committee (E587 and E632). Mice were housed under 12 hours light-dark cycles at 20-26 C and 20-70% humidity, in ventilated cages and constant access to food and water. The genetic *Pik3ca*[23] and *Grhl3*[24] mouse

**Fig. 6 | The PI3K-phospho-YBX1 axis in basal and mesenchymal HNC subtypes predicts patient prognosis. A** YBX1 expression in patients' HNC did not vary between basal ($n = 31$) and mesenchymal subtypes ($n = 27$) however, phospho-YBX1 was significantly higher in the basal group. Data are shown as mean ± SEM. The comparison between the two groups was performed using an unpaired two-tailed t test (* $p$-value = 0.0246). **B** The correlation of ssGSEA scores between the hallmark EMT gene set and YBX1 (R = 0.2187, two-sided $p$-value = 0.091) or phospho-YBX1 (R = −0.3438, $p$-value < 0.01) from basal and mesenchymal HNC ($n = 58$, Spearman's correlation). 95% confidence bands of the best-fit line are shown in pink. **C** Heatmap showing unsupervised clustering of HNC samples correlating levels of YBX1 and EMT ssGSEA with the basal subtype, and levels of phospho-YBX1 and PI3K/AKT/mTOR signalling ssGSEA with the mesenchymal subtype. **D** HR and adjusted Kaplan-Meier curve analyses of pYBX1 for overall survival of $n = 337$ patients from the TCGA-HNC cohort. HRs were estimated using multivariate Cox proportional hazard models. Asterisks indicate statistical significance per clinically-relevant factor. HR estimates are shown as dots at the centre of error bars and error bars

represent 95% confidence intervals of hazard ratios by two-sided Wald test. **E** Neoplasm histologic grade, neoplasm disease stage, and lymph node stage from the TCGA-HNC were compared for p-YBX1$^{Hi}$ and p-YBX1$^{Lo}$ HNC. $p$-value was calculated using Pearson's Chi-square test (* $p$-value = 0.0255, *** $p$-value = 0.0006, **** $p$-value < 0.0001). **F** YBX1 and phospho-YBX1 levels in the p-YBX1$^{Lo}$ ($n = 71$) and p-YBX1$^{Hi}$ ($n = 274$) patient groups. Data are shown as mean ± SEM. The comparison between the two groups was performed using an unpaired two-tailed t test (**** $p$-value < 0.0001). **G** Volcano plot of differentially expressed proteins between p-YBX1$^{Lo}$ ($n = 71$) and p-YBX1$^{Hi}$ ($n = 241$) using the TCGA-HNC RPPA data. Labelled proteins associated with the PI3K/mTOR signalling were upregulated in p-YBX1$^{Hi}$ tumours. Significant differentially expressed proteins with two-sided $q$-value < 0.05 are highlighted in pink, NS (not significant). **H** IHC staining for phospho-YBX1 and PDPN in patient HNC tissue arrays. Expression of p-YBX1 and PDPN was mutually exclusive in proliferative cells at the core of primary tumours and the invasive front. Source data are provided as a Source Data file.

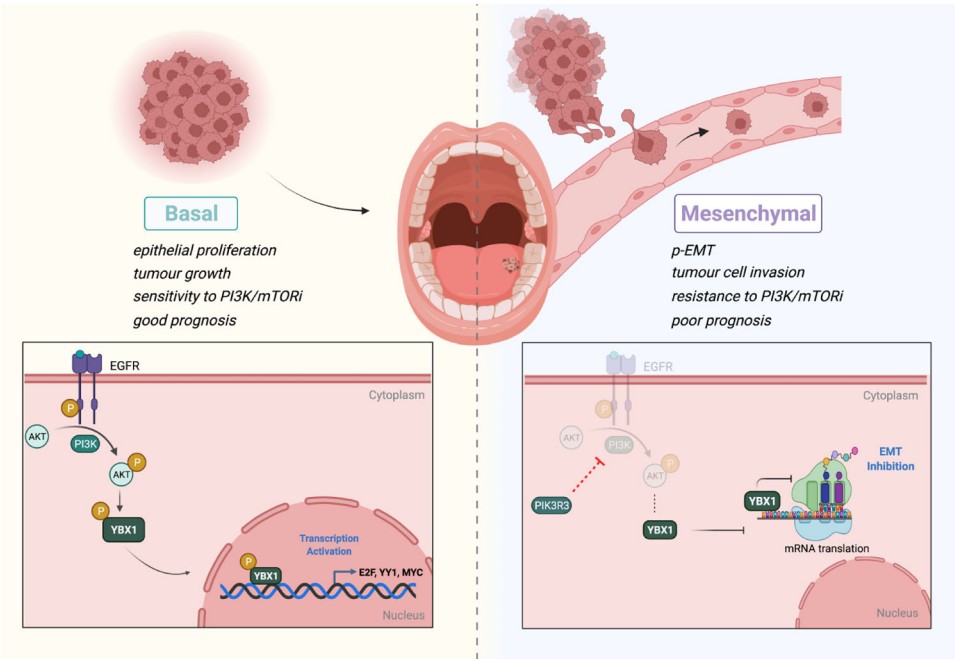

**Fig. 7 | Schematic diagram depicting the PI3K-phospho-YBX1 proliferation axis in HNC.** Oncogenic activation of PI3K signalling induces phosphorylation of YBX1 and nuclear translocation to promote cellular proliferation. Inactive PI3K signalling correlates with the cytoplasmic accumulation of YBX1 and the induction of EMT protein translation. The schematic was created using Biorender.com.

models were characterised previously. Both animal strains were established on a C57BL/6 background and males and females equally used at 3 months old. The animals were monitored according to the Peter Mac Animal Ethics guidelines and maximal tumour size and burden were not exceeded. The patient samples were collected under the University of Western Australia Human Research Ethics Approval number RA/4/1/8562 and made into tissue arrays. Informed consent was obtained by all participants. Established cell lines derived from patients with HPV-negative (SCC15 and SCC25) HNC were used to generate orthotopic xenografts in NSG mice (6–8 weeks of age). HNC cells ($n = 10,000$) were transduced with a lentivirus containing the luciferase (pCDH-EF1a-eFFly-mCherry, Addgene plasmid #104833) reporter and mCherry genes. The cells were then sorted and cultured to increase cell numbers. The cells were then resuspended in media containing Matrigel and culturing media (1:1) and orthotopically injected into the mouse tongue. Tumour growth was monitored weekly by bioluminescence imaging. The experimental mice were injected intraperitoneally with D-Luciferin (150 mg Luciferin/kg body weight prepared in PBS) and then anaesthetised using 4-unit

isoflurane. 5 minutes following luciferin injection, bioluminescence images were captured at 1 second exposure with the IVIS Spectrum In Vivo Imaging System. The mice were culled at the experimental endpoint, or earlier if they showed signs of distress, and their xenografts were collected for downstream analyses.

**Reverse phase protein array (RPPA)**
Protein lysates from human HNC cells were extracted using the RPPA lysis buffer and quantified by the Pierce Coomassie Plus (Bradford) Assay Kit (ThermoFisher Scientific, Cat No. 23236). Sample lysates were run at the Host and Tumour Profiling Unit (HTPU, Cancer Research UK, Edinburgh, UK). RPPA Relative Fluorescence Intensity (RFI) values were calculated by the Zeptoview software. A weighted linear regression through the dilution series was used to calculate the sample fluorescence intensity value which was then normalised to the reference BSA grid to account for intra-array spatial variation. Each intensity value was corrected to the background signal and the secondary antibody controls to validate the RFI value for each sample/antibody combination. '0' accounted for RFI values where the primary

antibody signal was lower than the signal emitted by the secondary antibody alone. 1e-9 was added to all RFI values for statistical analysis.

## Statistical analyses and reproducibility

Statistical significance was assessed using the unpaired Student's t test, one-way ANOVA, Spearman's correlation test, two-sided Wald test for multivariate Cox regression analysis or log-rank test for survival analysis using Prism 9 (GraphPad). Statistical analyses for the RPPA, RNA sequencing and single-cell RNA sequencing were carried out using R version 4.0.5. Packages used for single-cell RNA sequencing analysis included Cell Ranger pipeline v.3.0.2, Seurat_4.1.0; SummarizedExperiment_1.20.0; clustree_0.4.4; escape_1.3.3; SingleR_1.8.1. Packages used for RNA sequencing analysis included limma_3.46.0, GSEA_v4.3.2 and Galaxy suie environemtn (version 4.0, including Cutadpapt, HISAT2, featureCounts and Limma). Packages used for plotting included tidyverse_1.3.1, ComplexHeatmap_2.8.0 and ggpubr_0.4.0. The flow cytometry data was analyzed using FACSDiva 9.0. SPOT software 5.0, CellProfiler_4.2.4 and HALO quantitative image analysis platform 2.0 were used for IHC image analysis. The results are presented as mean ± SEM. The statistical values were considered significant at *$p$-value < 0.05, **$p$-value < 0.01, ***$p$-value < 0.001 and **** $p$-value < 0.0001. Experiments were repeated at least once. Replicates were reproducible. Schematic illustrations in Fig. 3E, Fig. 7 and supplementary Fig. 4B were created using Biorender.com.

## Reporting summary

Further information on research design is available in the Nature Portfolio Reporting Summary linked to this article.

## Data availability

The data that support this study are available in this paper and stored in GEO database as SuperSeries GSE226357 ("GSE226357"), including RNAseq raw data ("GSE226355"), and RPPA raw data ("GSE226356") or from the corresponding author (charbel.darido@petermac.org) upon reasonable request. Publicly available data from TCGA, Broad Institute and Stanford University were used. TCGA genetic, transcriptomic, proteomic, and clinical data were downloaded from the cBioportal data portal (https://www.cbioportal.org/; Head and Neck Squamous Cell Carcinoma (TCGA, PanCancer Atlas)). Genetic and transcriptomic data from Broad Institute were downloaded from depmap portal (https://depmap.org/portal/; CCLE_expression.csv; CCLE_mutation.csv). The single-cell RNA sequencing data ("GSE140042"), ("GSE103322"), ("GSE164690") were downloaded from GEO database (https://www.ncbi.nlm.nih.gov/gds). Raw data for bulk mRNA-seq on mouse tissues and human cancer cell lines and RPPA data from different conditions have been deposited on the Figshare repository. Raw data on mouse tissues and human cancer cell lines from different conditions are available on figshare; for bulk mRNA-seq ("20024246") and for RPPA ("20024258"). Public single-cell RNA-seq data on patient samples were downloaded from GSE140042, GSE103322 and GSE164690. The processed single-cell RNA-seq data is available on figshare ("20033024"). Additional data are available as supplementary materials and source data as Source Data files. Source data are provided with this paper.

## Code availability

All software algorithms used for analysis are available for download from public repositories. All code used to generate figures in the manuscript are made available in the following Github repository: https://github.com/DaridoLab-HNSC/YBX1-integration-of-oncogenic-PI3K-signalling.git.

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

## Acknowledgements

We thank members of the Darido laboratory including Imran Khan, Adelle Marrazzo, Sarah Fourari and Jade Tran for technical and general assistance with experiments and figures. The authors would like to acknowledge the assistance of David Goode from the Computational Cancer Biology Programme at the Peter MacCallum Cancer Centre. We also acknowledge the assistance from the Peter MacCallum Cancer Centre Core Facilities including the Bioinformatics Core Facility, the Victorian Centre for Functional Genomics, the Molecular Genomics Centre, the Centre for Advanced Histology & Microscopy and the Animal Facility. We thank Oliver Bissinger for the intellectual contribution and support of the co-author CG. This research was supported by a grant from the Australian National Health and Medical Research Council (NHMRC, APP1106697) to SJM and CD, and a Victorian Cancer Agency mid-career Fellowship (MCRF16017) to CD.

## Author contributions

Conceptualisation: Y.B., C.G., C.D.; Data curation: Y.B., L.F., C.D.; Formal analysis: Y.B., C.G., G.C., Z.Z., C.S., C.D.; Investigation: Y.B., C.G., G.C., Z.Z., J.B., C.A.; Methodology: Y.B., L.F., S.M.J., W.A.P., S.S., C.S.F., C.D.; Resources: C.S., L.F., S.M.J., W.A.P., S.S., C.S.F., C.D.; Supervision: C.S., C.S.F., C.D.; Writing–original draft: Y.B., C.G., C.D.; Writing–review and editing: Y.B., C.G., G.C., Z.Z., C.S., J.B., L.F., C.A., S.M.J., W.A.P., S.S., C.S.F., C.D.; Funding acquisition: S.M.J. and C.D. The funders had no role in the design of the study; the collection, analysis, or interpretation of the data; the writing of the manuscript; or the decision to submit the manuscript for publication.

## Competing interests

The authors declare no competing interests.
