## [Peer Review File · Nature Communications]

YBX1 integration of oncogenic PI3K/mTOR signalling regulates the fitness of malignant epithelial cellsREVIEWER COMMENTS

Reviewer #1 (Remarks to the Author):

This manuscript explores the relationship between PI3K-AKT-mTOR signaling and YBX1 regulation in cancer cell proliferation and invasion of HNSCC. The study is based on a broad range of timely analytical tools and bioinformatic workflows utilizing patient samples, including publicly available multi-omics data, as well as in vitro cell culture and in vivo mouse models. The manuscript is well-written and addressed an unmet medical need of high clinical relevance in the field of interest. However, additional experiments and controls are required to support final conclusions and improve the impact of the study prior publication in this journal.

Major points:

1. For Fig. 1 the Hallmark gene set from MSigDB was used to compute an EMT-related ssGSEA score for cases from TCGA-HNSC. Based on this score authors demonstrate a significantly higher EMT score in the mesenchymal as compared to basal subtype (Fig. 1C) and an inverse correlation with PI3K-AKT pathway activity (Fig. 1D-E). Both, the EMT-related ssGSEA score as well as the stratification of the mesenchymal subtype might be biased by variable stromal versus cancer cell ratios of the bulk RNA-seq data and authors should consider sample purity as a critical variable for data normalization.
2. Survival data are provided as KM plots (Fig. 1B, Fig. S1B, Fig. 6D) and are not adjusted for clinically-relevant risk factors or treatment (e.g. radiotherapy). Please add data from multivariable Cox regression models to confirm independent risk for unfavorable survival of the EMT high group or cases with low YBX1 phosphorylation. Also add the number at risk for indicated time points below each KM plot.
3. At page 6 lines 114-115 authors state that scRNA-seq profiles of primary and metastatic tumours were generated from four treatment-naïve patients. These data were retrieved from GSE140042 (page 17, line 366), which consists of 9 samples from 7 HNSCC patients in total. Why didn't the authors include all samples for this study and what were the selection criteria?
4. Fig. 2H demonstrates an enrichment for the EMT signature in G1-arrested cancer cells while PI3K-AKT-mTOR signaling was active in cycling cells (S and G2/M phases) for malignant epithelial cells. Though significant presented differences are minor and should be confirmed by independent scRNA-seq data (PMID: 29198524, PMID: 34921143). Validation with more cells from independent scRNA-seq data sets would also be worth for data shown in Fig. 2K as significant differences are only detected for one sample from metastasis and primary tumor, respectively.
5. Data from the mouse model presented in Fig. S3 remain descriptive and the added value to support final conclusions is vague. Is PI3K hyperactivation associated with YBX1 phosphorylation in this model, and are those tumors sensitive to BEZ235 inhibitor treatment? Do animals with spontaneous tumors lack lymph node or distant metastasis even after a longer follow-up period? Concerning clinical relevance: Are basal type HNSCC with PI3K pathway activity and YBX1 phosphorylation from TCGA-HNSC characterized by low or absent GRHL3 expression?
6. SCC15 and SCC25 resembled a mesenchymal or basal subtype with variable PI3K pathway activity and were selected for further analysis. Does inhibition of PI3K pathway activity by BEZ235 reduce YBX1 phosphorylation and nuclear-cytoplasmic translocation and induce a p-EMT phenotype in SCC25 cells in vitro or in vivo?
7. At page 8 lines 172-175 authors report a difference in regional lymph node metastasis between orthotopic xenograft mouse models generated with either SCC15 or SCC25 cells, which is not supported by quantitative data and statistical evaluation. Was the presence of lymph node metastasis independent on primary tumor size and did authors also detect distant (e.g. lung) metastasis? Same issue with statement at page 12 lines 246-248 as Fig. 5F does not show any quantitative data on significant differences in metastatic potential.
8. Fig. 4A demonstrates a 50% knockout efficacy for YBX1 in SCC25 cells, but residual tumors were negative by IHC (Fig. 4H). In support of their conclusion that YBX1 expression and phosphorylation promotes tumor growth of basal subtypes, one would expect a positive selection of SCC25 cells with inefficient silencing and residual YBX1 expression in vivo over time. Moreover, the significant difference between SCC25 with or without YBX1 knockout around 4-weeks post orthotopic implantation (Fig. 4G) is not present in Fig. S5 prior initiation of BEZ235 treatment. Do mouse xenografts with SCC25-YBX1 cells show lymph node or distant metastasis?
9. All animal studies with the PI3K-mTOR inhibitor BEZ235 (Fig. S5) lack a proper control group

with untreated animals, which hampers adequate assessment of treatment efficacy (e.g. IHC staining of downstream targets with phosphorylation specific antibodies) and treatment response. 10. Fig. 5G should demonstrate YBX1 expression at the invasive front of primary tumors. However, positive staining appears to be nuclear and more likely in mouse stromal rather than human cancer cells. As virus-transduced cancer cells do not only express luciferase but also mCherry confirmation of YBX1 expression in SCC15+YBX1 cells could easily be done by co-IF analysis with cryosections.

11. Fig. 6H shows representative images of an IHC staining for three HNSCC samples from a tissue microarray. How many samples were present on the TMA in total? In case an adequate number of samples/patients is available, authors should consider a (semi-)quantitative assessment to further confirm the clinical relevance (Fig. 6D-E) with an independent cohort.

Minor points:

1. IHC staining shown in Fig. 3F is of poor quality. Please show representative pictures at lower magnification and provide more detailed information in Materials and Methods, how specificity of antibody staining was confirmed. Same concern for Fig. 4H and Fig. 5G.
2. Please provide the total number of animals per group, which were analyzed and used for statistical evaluation in Fig. 3, Fig. 5 and Fig. S5.

Reviewer #2 (Remarks to the Author):

This paper elucidates the dual role of PI3K-dependent phosphorylation of YBX1 in promoting partial epithelial to mesenchymal transition in two different subtypes of head and neck cancer (HNC). Using bioinformatic, genetic, and pharmacological approaches as well as clinical data, the paper concludes that phosphorylated YBX1 protects HNC switch from just proliferation to invasion. As such, the data in this paper provides nuances that should be considered in the molecular signature of HNC patients for effective treatment purposes.

Suggestions

1. Fig.1 Text size is small in figure 1.
2. Fig. 3H: The magnifications of top panel of figure 3H and bottom panel are different. It looks like only YBX1 panel is 40X and p-YBX1 is at a lower magnification. Fig. 3H quantification of IF could be useful. SCC25 figure is a little blurry.
3. Figure 4C and 5C are also hard to see at a distance.
4. Suggest to add a clear model in the main figures instead of in the supplementary information.
5. Figure S3C: It is claimed there is no p-EMT due a lack of PDPN. Is this the only marker of p-EMT used?
6. Figure legend S4C, concludes that PI3K activation by EGF and FBS induces p-YBX1 in SCC25 but not SCC15. However, the western blot presented showed slight increase in phosphorylated YBX1 under EGF and FBS treatment in SCC15. The description of the figure should acknowledge this slight increase or show a quantification for statistical significance purposes if current figure legend description is correctly stated.
7. Line 54-55 of the supplementary figure concluded that EGF treatment promoted nuclear localization of p-YBX1 in SCC25. However, the western blot shown in figure S4E demonstrates no change in cytoplasmic levels versus the nuclear levels of p-YBX1. Given that total YBX1 in the nuclear fraction of SCC25 appears lower than that of the cytoplasmic fraction, I suggest presenting the quantification of the nuclear to cytoplasmic ratio of p-YBX1/total YBX1.
8. In the main text, line 313 to 315 suggests that total YBX1 is similar across all HNC subtypes. However, the data presented in this paper showed otherwise; particularly in the cell lines SCC15 and SCC25, which represents mesenchymal and basal HNC, respectively.
9. The use of one "basal" and one "mesenchymal" cell seems appropriate but "atypical" and "classical" types are mentioned and never used. This could possibly be justified more in the text.

Minor concerns with some word choice:

Inverse correlation is used frequently. Maybe other words could be used to describe this? First three figures have very similar titles because of this.

Reviewer #3 (Remarks to the Author):

The manuscript entitled "YBX1 integration of oncogenic PI3K/mTOR signaling regulates the fitness of malignant epithelial cells" describes the involvement of YB-1 in proliferation and EMT, which apparently depends on the PI3K/mTOR activity in HNC

The manuscript is well written and easy to follow. Sophisticated in vivo approaches have been performed and clinical data have been analyzed, too. Together they support the established hypothesis which is summarized nicely in figure S6B. Although the translational impact has been discussed, this remains unclear/unsolved.

Specific comments

1. Figure 3H. The quality of the immune staining is unsatisfactory and must be repeated. Please do not show overlay YB-1/DAPI only. See please also Sogorina et al Int. J Molecular Sciences 2021 showing IF pictures of high quality
2. It is well established that cells showing a characteristic EMT phenotype display a drug resistant phenotype. In addition, chemotherapy is one standard treatment option for HNC. Therefore providing information regarding IC-50 values of Cisplatin/5-Fu of the used cell lines are important
3. In the discussion part the induction of EMT in Cetuximab treated HNC in the clinic was mentioned. In this regard, similar to question 2 it would be valuable to address the question for the used cell lines. Does the used cell lines response differently to Cetuximab ?
4. Line 1174-1176: Statement that in the PI3K-inactive mesenchymal subtype confers a metastatic potential in vivo are not shown completely. Please provide data regarding metastasis to the lymph nodes.
5. Lane 198-199 and suppl. : „EGF treatment promoted the nuclear localization of p-YBX1 in SCC25 but not in SCC15 cells.“ This statement is not supported convincingly by the WBA in figure 4SE. Please check and update a new figure
6. In the discussion part, I am missing a comment regarding the current treatment modality with cisplatin/5-FU. Should we treat all patient with chemotherapy equally?
7. Experiments using siRNA have been presented. However, I am missing experiments showing overexpression of YB-1 (ore an YB-1 phosphorylation mimic), which causes nuclear localization. It would be interesting to evaluate if overexpression of YB-1 can reverse the EMT phenotype.
8. Please include the expression of the mesenchymal markers N-cadherin, Snail, Twist and vimentin, and the epithelial marker E-cadherin. WBA ore IF studies for the main cell lines used SCC15 and SCC25

Response to Reviewers

Reviewer #1 (Remarks to the Author):

This manuscript explores the relationship between PI3K-AKT-mTOR signaling and YBX1 regulation in cancer cell proliferation and invasion of HNSCC. The study is based on a broad range of timely analytical tools and bioinformatic workflows utilizing patient samples, including publicly available multi-omics data, as well as in vitro cell culture and in vivo mouse models. The manuscript is well-written and addressed an unmet medical need of high clinical relevance in the field of interest. However, additional experiments and controls are required to support final conclusions and improve the impact of the study prior publication in this journal. We are very thankful to the Reviewer for assessing our manuscript and for providing the work with constructive feedback.

Major points:

1. For Fig. 1 the Hallmark gene set from MSigDB was used to compute an EMT-related ssGSEA score for cases from TCGA-HNSC. Based on this score authors demonstrate a significantly higher EMT score in the mesenchymal as compared to basal subtype (Fig. 1C) and an inverse correlation with PI3K-AKT pathway activity (Fig. 1D-E). Both, the EMT-related ssGSEA score as well as the stratification of the mesenchymal subtype might be biased by variable stromal versus cancer cell ratios of the bulk RNA-seq data and authors should consider sample purity as a critical variable for data normalization.

We thank the Reviewer and agree with them on the inclusion of sample purity. The ESTIMATE and Stroma scores were performed on bulk RNA-seq and show a biased EMT signature in the mesenchymal subtype as shown below. This data is now included as Fig. S1B in the supplemental material which confirms the limitation of bulk RNA-seq in Fig. 1 and further justifies the epithelial-specific single-cell analysis in Fig. 2.

2. Survival data are provided as KM plots (Fig. 1B, Fig. S1B, Fig. 6D) and are not adjusted for clinically-relevant risk factors or treatment (e.g. radiotherapy). Please add data from multivariable Cox regression models to confirm independent risk for unfavorable survival of the EMT high group or cases with low YBX1 phosphorylation. Also add the number at risk for indicated time points below each KM plot.

We thank the Reviewer for this important and constructive comment. Using multivariable Cox regression models for clinically-relevant risk factors (subtype, age at diagnosis, sex) and treatment (radiation therapy), statistically significant data confirm the independent risk for unfavorable survival of patients from the EMT^{hi} or low phospho-YBX1 groups. This is now added as main Figures 1B and 6D and supplemental Figure S1C. The number at risk for the indicated time points is shown below Fig. 1B and Fig. 6D.

3. At page 6 lines 114-115 authors state that scRNA-seq profiles of primary and metastatic tumours were generated from four treatment-naïve patients. These data were retrieved from GSE140042 (page 17, line 366), which consists of 9 samples from 7 HNSCC patients in total. Why didn't the authors include all samples for this study and what were the selection criteria? We appreciate the Reviewer's precise comment. Only samples from GSE140042 with 2,000 to 4,000 sequenced cells were considered for downstream analyses while others were excluded. Cells with more than 200 RNA features were retained and RNA features detected in more than 2 cells were considered for scRNA-seq analyses. These criteria allowed selection of six primary and metastatic tumours (from four treatment-naïve patients) after initial quality controls and partitioned into 13 clusters by gene expression levels (Fig. 2). This information was in the reporting summary at submission and is now included in the Materials and Methods section.

4. Fig. 2H demonstrates an enrichment for the EMT signature in G1-arrested cancer cells while PI3K-AKT-mTOR signaling was active in cycling cells (S and G2/M phases) for malignant epithelial cells. Though significant presented differences are minor and should be confirmed by independent scRNA-seq data (PMID: 29198524, PMID: 34921143). Validation with more

Rebuttal Letter

cells from independent scRNA-seq data sets would also be worth for data shown in Fig. 2K as significant differences are only detected for one sample from metastasis and primary tumor, respectively. Adding analysis using other external datasets (PMID: 29198524; PMID: 34921143)

We thank the Reviewer for bringing this up. We have analysed the external datasets GSE103322 and GSE164690 as suggested. The cell cycle analysis of GSE103322 shows enrichment for the EMT signature in G1-arrested cancer cells while PI3K-AKT-mTOR signalling was active in cycling cells in a similar trend to our results (Figure S2A), and the scRNA-seq of GSE103322 in Puram et al., 2017 clearly confirms our finding. The analysis of GSE164690 also shows significant results that confirm our data (Figures S2B and S2C). These analyses are now presented as a new Supplemental Figure S2.

5. Data from the mouse model presented in Fig. S3 remain descriptive and the added value to support final conclusions is vague. Is PI3K hyperactivation associated with YBX1 phosphorylation in this model, and are those tumors sensitive to BEZ235 inhibitor treatment? Do animals with spontaneous tumors lack lymph node or distant metastasis even after a longer follow-up period? Concerning clinical relevance: Are basal type HNSCC with PI3K pathway activity and YBX1 phosphorylation from TCGA-HNSC characterized by low or absent GRHL3 expression?

We thank and agree with the Reviewer. We have now included an immunohistochemistry staining that supports increased YBX1 phosphorylation in tumours from double mutant *Pik3ca*^{H1047R} *Grhl3*^{CKO} mice compared to WT and single mutant animals in Figure S4D. For the sensitivity of those tumours to BEZ235 treatment, we elected to treat primary tumour cells *in vitro* instead of the *Pik3ca*^{H1047R} *Grhl3*^{CKO} mice as the breeding, ageing and *in vivo* treatment for this experiment could take over 6 months. *Pik3ca*^{H1047R} *Grhl3*^{CKO}-derived tumour cells exhibit a low IC₅₀ of 0.886 nM, indicating that those tumours will be sensitive to BEZ235 treatment. Additionally, SCC25 cells (active PI3K) are more sensitive to BEZ235 with an IC₅₀ = 69.3 nM compared to SCC15 (inactive PI3K) with an IC₅₀ = 119.4 nM.

The longest follow-up period of these animals was 6 months as they were losing weight and had to be culled in accordance with our animal ethics guidelines, preventing further ageing to assess for lymph node and distant metastasis. Post mortem examination at 6 months did not show signs of lymph node or distant metastasis in tumour-bearing *Pik3ca*^{H1047R} *Grhl3*^{CKO} mice. With regards to clinical relevance, HNSCC of the basal subtype from the TCGA-HNC are characterised by low GRHL3 expression compared to control adjacent tissues. This data is added in Figure S3 now becoming Figure S4A.

Fig. S4D

Fig. S4A

6. SCC15 and SCC25 resembled a mesenchymal or basal subtype with variable PI3K pathway activity and were selected for further analysis. Does inhibition of PI3K pathway activity by BEZ235 reduce YBX1 phosphorylation and nuclear-cytoplasmic translocation and induce a p-EMT phenotype in SCC25 cells *in vitro* or *in vivo*?

We thank the Reviewer on this question. We now present data on SCC25 cells following BEZ235 inhibition of the PI3K pathway activity. Western blotting experiments on cell fractions of SCC25-treated cells *in vitro* show reduced YBX1 phosphorylation in both the nuclear and cytoplasmic fractions. BEZ235 treatment of SCC25 cells did not induce a p-EMT phenotype. This is shown by the absence of PDPN induction or CDH1 loss of expression in BEZ235-treated SCC25 cells.

7. At page 8 lines 172-175 authors report a difference in regional lymph node metastasis between orthotopic xenograft mouse models generated with either SCC15 or SCC25 cells, which is not supported by quantitative data and statistical evaluation. Was the presence of lymph node metastasis independent on primary tumor size and did authors also detect distant (e.g. lung) metastasis? Same issue with statement at page 12 lines 246-248 as Fig. 5F does not show any quantitative data on significant differences in metastatic potential. We thank the Reviewer on this question. The regional lymph node metastasis was detected for SCC15 cells only even though SCC25 cells grew bigger primary tumours (Figure S5C, left panel). Distant metastasis (e.g. to lung) in SCC15-grafted animals were not detected within the period of analysis (Figure S5C, table) and this may account to the early culling of the mice that showed health deterioration in accordance with our animal ethics guidelines. There was no significant difference in primary SCC15 +YBX1 and SCC15 -YBX1 tumour volume (Figure S5C, right panel) indicating that the tumour size did not correlate with regional lymph node metastasis.

8. Fig. 4A demonstrates a 50% knockout efficacy for YBX1 in SCC25 cells, but residual tumors were negative by IHC (Fig. 4H). In support of their conclusion that YBX1 expression and phosphorylation promotes tumor growth of basal subtypes, one would expect a positive selection of SCC25 cells with inefficient silencing and residual YBX1 expression *in vivo* over time. Moreover, the significant difference between SCC25 with or without YBX1 knockout around 4-weeks post orthotopic implantation (Fig. 4G) is not present in Fig. S5 prior initiation

of BEZ235 treatment. Do mouse xenografts with SCC25-YBX1 cells show lymph node or distant metastasis?

Thank you for this interesting comment. SCC25 cells have lower YBX1 expression compared to SCC15 (Figure 3G). We have now quantified YBX1 protein expression in the western blot of Fig. 4A to show that the YBX1 protein level is reduced to 20% in SCC25 knockout cells. The low YBX1 expression was not detectable by IHC in Fig. 4H indicating that cells with residual YBX1 did not have a growth advantage.

We have included the data on SCC25 with or without YBX1 in the absence of BEZ235 treatment in Fig. S5 now becoming Fig. S6, please see our response to point 9.

Same as point 7, the mouse xenografts with SCC25 -YBX1 cells did not show lymph node or distant metastasis. This is in agreement with BEZ235 treatment that did not induce a p-EMT phenotype in SCC25 (point 6).

9. All animal studies with the PI3K-mTOR inhibitor BEZ235 (Fig. S5) lack a proper control group with untreated animals, which hampers adequate assessment of treatment efficacy (e.g. IHC staining of downstream targets with phosphorylation specific antibodies) and treatment response.

We agreed with the Reviewer and repeated this experiment with the inclusion of untreated animals (Figures S6A and S6B) and assessed the treatment efficacy of BEZ235 by IHC staining for phospho-RPS6 (and total RPS6) (Figure S6C).

10. Fig. 5G should demonstrate YBX1 expression at the invasive front of primary tumors. However, positive staining appears to be nuclear and more likely in mouse stromal rather than human cancer cells. As virus-transduced cancer cells do not only express luciferase but also mCherry confirmation of YBX1 expression in SCC15+YBX1 cells could easily be done by co-IF analysis with cryosections.

We thank the Reviewer on this comment. We have performed a co-IF analysis using anti-GFP (Cas9 is GFP-tagged) and anti-YBX1 (red) on our SCC15 +YBX1 sections. The data shows cytoplasmic expression of YBX1 in GFP-positive human cells at the invasive front of primary tumours. DAPI shows mouse stromal cells between dashed lines that are negative for GFP and YBX1. We have included the merged co-IF (GFP/YBX1/DAPI) in Figure 5G.

11. Fig. 6H shows representative images of an IHC staining for three HNSCC samples from a tissue microarray. How many samples were present on the TMA in total? In case an adequate number of samples/patients is available, authors should consider a (semi-)quantitative assessment to further confirm the clinical relevance (Fig. 6D-E) with an independent cohort. We appreciate the Reviewer’s question. The patient samples (n=135) were collected for another study and their de-identified data did not include clinical information and survival status. We are providing a summary table with the available pathology and molecular characteristics for each tumour. The HNSCC tumours are graded as well, moderately or poorly differentiated by our pathologist. The Ki67 readouts are 0, 1, 2, 3, 4 indicating Zero, 1-25%, 26-50%, 51-75%, 76-100% nuclear staining. The p16 readouts are negative or positive (must have had at least 75% nuclear and cytoplasmic positivity to be deemed positive). The p53 readouts are negative, positive and overexpressed. p53 positive indicates low level normal staining. p53 overexpression is scored for beyond positive expression across the full sample. We have quantified the staining intensity for PDPN and pYBX1 in each of the samples using the Halo software for quantitative image analysis of the selected ROI: Region Of Interest in serial sections. The statistical analysis indicates a significant inversed correlation between PDPN and pYBX1 across the 135 patient samples. This data is now presented in Figure S8A-C.

	Patient No. (n=135)
Differentiation Grading	
Well	51
Moderate	60
Poor	23
Ki67	
0	13
1	67
2	37
3	6
4	12
p16	
Negative	111
Positive	24
p53	
Negative	68
Positive	48
Overexpression	19

Minor points:

1. IHC staining shown in Fig. 3F is of poor quality. Please show representative pictures at lower magnification and provide more detailed information in Materials and Methods, how specificity of antibody staining was confirmed. Same concern for Fig. 4H and Fig. 5G.

We replaced the IHC for PDPN and CDH1 in Figures 3F, 4H and 5G with better quality pictures and included lower magnification (20X) images to the Reviewer. No primary or no secondary antibodies were included in our IHC experiments as controls for the antibody staining specificity. Knockout cells were additional negative controls to validate the YBX1 antibody specificity. This information is now added to the Materials and Methods section.

2. Please provide the total number of animals per group, which were analyzed and used for statistical evaluation in Fig. 3, Fig. 5 and Fig. S5.

The number of animals analysed and used for statistical evaluation was as follow:

Figure 3E, n= 12 mice per group.

Figure 4G, n= 9 mice per group.

Figure 5F, n= 9 mice per group.

Figure S6A (previously S5), n= 6 mice per group.

This information is now added to each figure legend. Thank you.

Reviewer #2 (Remarks to the Author):

This paper elucidates the dual role of PI3K-dependent phosphorylation of YBX1 in promoting partial epithelial to mesenchymal transition in two different subtypes of head and neck cancer (HNC). Using bioinformatic, genetic, and pharmacological approaches as well as clinical data, the paper concludes that phosphorylated YBX1 protects HNC switch from just proliferation to invasion. As such, the data in this paper provides nuances that should be considered in the molecular signature of HNC patients for effective treatment purposes.

We are very thankful to the Reviewer for assessing our manuscript and for providing the work with constructive suggestions.

Suggestions

1. Fig.1 Text size is small in figure 1.

We thank the Reviewer for pointing this out. We have now increased the text size in figure 1 where possible, particularly in Figure 1E.

2. Fig. 3H: The magnifications of top panel of figure 3H and bottom panel are different. It looks like only YBX1 panel is 40X and p-YBX1 is at a lower magnification. Fig. 3H quantification of IF could be useful. SCC25 figure is a little blurry.

We thank the Reviewer for this suggestion. We have now replaced the pictures in Figure 3H with the same magnifications and provided quantification of the nuclear/cytoplasmic fluorescence for YBX1 and the nuclear fluorescence intensity for pYBX1.

3. Figure 4C and 5C are also hard to see at a distance.

Thank you. We have now replaced Figures 4C and 5C with higher magnification pictures.

4. Suggest adding a clear model in the main figures instead of in the supplementary information.

As suggested, we have moved the schematic representation of the models to the main figures as Figure 7.

5. Figure S3C: It is claimed there is no p-EMT due a lack of PDPN. Is this the only marker of p-EMT used?

We thank the Reviewer for this question. We have used other p-EMT markers such as Vimentin that is also absent in epithelial cells, in a similar fashion to PDPN. Additionally, we have performed an IHC for pYBX1 which shows expression of YBX1 in epithelial cells of *Pik3ca*^{H1047R}*Grhl3*^{cKO} tumours in the absence of p-EMT (Figure S4D).

6. Figure legend S4C, concludes that PI3K activation by EGF and FBS induces p-YBX1 in SCC25 but not SCC15. However, the western blot presented showed slight increase in phosphorylated YBX1 under EGF and FBS treatment in SCC15. The description of the figure should acknowledge this slight increase or show a quantification for statistical significance purposes if current figure legend description is correctly stated.

Thank you for this comment. We have quantified the western blots in Figure S4C which is now Figure S5E and have amended the figure legend to acknowledge the slight increase of p-YBX1 in SCC15 under EGF and FBS treatment.

7. Line 54-55 of the supplementary figure concluded that EGF treatment promoted nuclear localization of p-YBX1 in SCC25. However, the western blot shown in figure S4E demonstrates no change in cytoplasmic levels versus the nuclear levels of p-YBX1. Given that total YBX1 in the nuclear fraction of SCC25 appears lower than that of the cytoplasmic fraction, I suggest presenting the quantification of the nuclear to cytoplasmic ratio of p-YBX1/total YBX1.

Thank you for this great suggestion. We have quantified the p-YBX1 and YBX1 western blots in Figure S4E which is now Figure S5G and presented the nuclear to cytoplasmic ratio of p-YBX1/YBX1.

8. In the main text, line 313 to 315 suggests that total YBX1 is similar across all HNC subtypes. However, the data presented in this paper showed otherwise; particularly in the cell lines SCC15 and SCC25, which represents mesenchymal and basal HNC, respectively.

We agree with the Reviewer. We have amended the mentioned lines (now line 388) to keep them consistent with our data and discussion in Line 217 to 219 “WB analyses showed the highest level of total YBX1 in mesenchymal SCC15 cells and phosphorylated forms of YBX1 in basal SCC25 cells within the HNC cell lines (Figure 3G)”.

Rebuttal Letter

9. The use of one "basal" and one "mesenchymal" cell seems appropriate but "atypical" and "classical" types are mentioned and never used. This could possibly be justified more in the text.

Thank you. As suggested, we have added additional information in the introduction (lines 52-53) to justify our focus on the basal and mesenchymal types.

Minor concerns with some word choice:

Inverse correlation is used frequently. Maybe other words could be used to describe this? First three figures have very similar titles because of this.

We have amended the titles of the first three figures as suggested. Thank you.

Reviewer #3 (Remarks to the Author):

The manuscript entitled “YBX1 integration of oncogenic PI3K/mTOR signaling regulates the fitness of malignant epithelial cells” describes the involvement of YB-1 in proliferation and EMT, which apparently depends on the PI3K/mTOR activity in HNC

The manuscript is well written and easy to follow. Sophisticated in vivo approaches have been performed and clinical data have been analyzed, too. Together they support the established hypothesis which is summarized nicely in figure S6B. Although the translational impact has been discussed, this remains unclear/unsolved.

We are very thankful to the Reviewer for assessing our manuscript and for providing the work with constructive feedback.

Specific comments

1. Figure 3H. The quality of the immune staining is unsatisfactory and must be repeated. Please do not show overlay YB-1/DAPI only. See please also Sogorina et al Int. J Molecular Sciences 2021 showing IF pictures of high quality IF needs to be redone and imaged using confocal microscope.

We thank the Reviewer for this comment. We have now replaced the IF pictures in Figure 3H with high quality confocal microscopy images and provided quantification of the nuclear/cytoplasmic fluorescence for YBX1 and the nuclear fluorescence intensity for pYBX1. Please see our response to Reviewer 2, point 2.

2. It is well established that cells showing a characteristic EMT phenotype display a drug resistant phenotype. In addition, chemotherapy is one standard treatment option for HNC. Therefore providing information regarding IC-50 values of Cisplatin/5-Fu of the used cell lines are important. We thank the Reviewer for this important point. In searching the literature, we found references to IC-50 values for Cisplatin (Lima et al., 2022) and IC-50 values for 5-FU (Chen et al., 2018) that validate our findings in SCC15 and SCC25 with the mesenchymal SCC15 cells being more resistant to chemotherapy. Both references have been included in the manuscript.

3. In the discussion part the induction of EMT in Cetuximab treated HNC in the clinic was mentioned. In this regard, similar to question 2 it would be valuable to address the question for the used cell lines. Does the used cell lines response differently to Cetuximab ?

Thank you again for this comment. SCC15 and SCC25 have been shown to respond differently to Cetuximab with SCC15 being less responsive than SCC25 as shown in Kjaer et al., 2016. This reference has also been cited in the discussion.

4. Line 1174-1176: Statement that in the PI3K-inactive mesenchymal subtype confers a metastatic potential in vivo are not shown completely. Please provide data regarding metastasis to the lymph nodes.

We thank the Reviewer for this comment. We are now providing additional data in Figure S5B showing regional lymph node metastasis for the PI3K-inactive mesenchymal subtype (SCC15). We also provide a table summarising the number of metastasis detected in the lymph nodes. There was no significant difference in primary SCC15 +YBX1 and SCC15 –YBX1 tumour volume (Figure S5C, right panel) indicating that the tumour size did not correlate with regional lymph node metastasis. Please see our response to Reviewer 1, point 7.

5. Lane 198-199 and suppl. : „EGF treatment promoted the nuclear localization of p-YBX1 in

Rebuttal Letter

SCC25 but not in SCC15 cells.” This statement is not supported convincingly by the WBA in figure 4SE. Please check and update a new figure

Thank you for this great suggestion. We have updated Figure S4E which is now Figure S5G and quantified the p-YBX1 and YBX1 western blots and presented the nuclear to cytoplasmic ratio of p-YBX1/YBX1. Please see our response to Reviewer 2, point 7.

6. In the discussion part, I am missing a comment regarding the current treatment modality with cisplatin/5-FU. Should we treat all patient with chemotherapy equally?

Considering the differential response of basal and mesenchymal subtypes to Cisplatin/5-FU, HNC patient of the basal subtype are proposed to respond better than those of the mesenchymal subtype. The discussion has been updated accordingly (line 377 to 379).

7. Experiments using siRNA have been presented. However, I am missing experiments showing overexpression of YB-1 (ore an YB-1 phosphorylation mimic), which causes nuclear localization. It would be interesting to evaluate if overexpression of YB-1 can reverse the EMT phenotype.

We thank the Reviewer for this interesting comment. We have generated a YBX1 myc-tagged expressing construct for the overexpression (OE) of YBX1 in mesenchymal SCC15 cells. Transfected cells (57.4%) were confirmed for YBX1 OE using flow cytometry (Figure S7A) and by anti-MYC and anti-YBX1 western blots (Figure S7B). The transwell invasion assay shows OE cells with nuclear YBX1 expression in green (IF for the MYC tag) are retained in the inner membrane (non-invasive) whether SCC15 cells with cytoplasmic YBX1 are detected in the outer membrane (invasive). The quantification of YBX1 OE demonstrates decreased nuclear to cytoplasmic ratio in invasive cells. While YBX1 is mainly shown in the cytoplasm of SCC15 cells (Figure 3H) and response to Reviewer 2, point 2, this data indicates that OE of YBX1 in SCC15 can localise to the nucleus to inhibit cell invasion (Figure S7C).

8. Please include the expression of the mesenchymal markers N-cadherin, Snail, Twist and vimentin, and the epithelial marker E-cadherin. WBA ore IF studies for the main cell lines used SCC15 and SCC25

We have performed WB analyses with the suggested EMT markers on SCC15 and SCC25 cells. We found that CDH1 (E-cad) and CDH2 (N-cad) have opposite expression pattern in the cell lines. Vimentin, Snail and Twist1 are expressed at lower level in basal SCC25 compared to mesenchymal SCC15 cells. This data is now added as Figure S5B.

Thank you.

REVIEWERS' COMMENTS

Reviewer #1 (Remarks to the Author):

Authors adequately answered most questions and concerns, which were raised by the reviewers to improve the quality of the manuscript and to further support final conclusions of the study. However, following minor issues need to be addressed prior publication:

1. Multivariate Cox regression analysis was done to demonstrate that either the EMT signature (Fig. 1B) or YBX1 phosphorylation (pYBX1-S102, Fig. 6D) serve as independent risk factors for overall survival (OS) after adjustment for HPV status, age, sex and radiotherapy. Are selected variables the only risk factors with a significant impact on OS for the analyzed patient cohorts? Did the authors formally exclude any prognostic impact of other clinical variables, such as pathological tumor size, pathological lymph node metastasis and resection margins, on OS?

2. In Fig. S4A, GRHL3 transcript levels for primary tumors from non-basal subtypes should be added to demonstrate significantly lower expression in basal type HNSCC with PI3K activity and not only compared to adjacent normal tissue.

Reviewer #2 (Remarks to the Author):

The authors have fully addressed the previous critiques.

Reviewer #3 (Remarks to the Author):

My concerns have been addressed in the revisions satisfactorily

I have no objections regarding publishing this manuscript.

REVIEWERS' COMMENTS

Reviewer #1 (Remarks to the Author):

Authors adequately answered most questions and concerns, which were raised by the reviewers to improve the quality of the manuscript and to further support final conclusions of the study. However, following minor issues need to be addressed prior publication:

1. Multivariate Cox regression analysis was done to demonstrate that either the EMT signature (Fig. 1B) or YBX1 phosphorylation (pYBX1-S102, Fig. 6D) serve as independent risk factors for overall survival (OS) after adjustment for HPV status, age, sex and radiotherapy. Are selected variables the only risk factors with a significant impact on OS for the analyzed patient cohorts? We thank the Reviewer again for their thoughtful questions. The EMT signature or pYBX1-S102 serve as independent risk factors after adjustment to all risk factors included in the TCGA-HNC patient cohort (TCGA, Nature 2015). HPV, age, sex and radiotherapy are the only important risk factors on clinical grounds with a significant impact on OS. The survival analysis did not show a significant difference between smokers and non-smokers (Ghasemi et al., *JCI Insight* 2019). We speculate that this observation may have resulted from the heterogeneity of treatment effects within this patient cohort, which could obscure small differences in survival.

Did the authors formally exclude any prognostic impact of other clinical variables, such as pathological tumor size, pathological lymph node metastasis and resection margins, on OS? The number of patients analysed for OS was n=71 for phospho-YBX1^{low} and n=266 for phospho-YBX1^{hi}. As requested by the reviewer, we performed a survival analysis also adjusting by tumour grade, stage, and lymph node metastasis. No TCGA data is available on the resection margins. The addition of those clinical variables to our model did not impact on OS by Multivariate Cox regression analysis and therefore, the model presented in the manuscript does not include other clinical variables. Alternatively, using Pearson's Chi-square test, statistically significant differences were obtained when classifying the patients into tumour grade, stage, and lymph node metastasis between phospho-YBX1^{low} and phospho-YBX1^{hi} (Fig. 6E).

2. In Fig. S4A, GRHL3 transcript levels for primary tumors from non-basal subtypes should be added to demonstrate significantly lower expression in basal type HNSCC with PI3K activity and not only compared to adjacent normal tissue.

We thank the Reviewer again and as suggested have added the GRHL3 transcript levels for primary TCGA tumours from non-basal subtypes. We further averaged the GRHL3 levels from the 3 datasets of scRNA-seq (GSE103322, GSE164690 and GSE140042). While GRHL3 expression is low in all HNC patient samples, its level is comparable in clusters of epithelial cells of the basal subtype compared to non-basal subtypes. This data emphasises the role of PI3K/mTOR activity in regulating the malignant epithelial cell fitness in specific HNC subtypes, independent of how low is the GRHL3 level. Fig. S4A has been updated accordingly.

Reviewer #2 (Remarks to the Author):

The authors have fully addressed the previous critiques.

Reviewer #3 (Remarks to the Author):

My concerns have been addressed in the revisions satisfactorily.
I have no objections regarding publishing this manuscript.